# A shared neural code for the physics of actions and object events

Seda Karakose-Akbiyik [1] ✉, Alfonso Caramazza[1,2] & Moritz F. Wurm[2]

Observing others' actions recruits frontoparietal and posterior temporal brain regions – also called the action observation network. It is typically assumed that these regions support recognizing actions of animate entities (e.g., person jumping over a box). However, objects can also participate in events with rich meaning and structure (e.g., ball bouncing over a box). So far, it has not been clarified which brain regions encode information specific to goal-directed actions or more general information that also defines object events. Here, we show a shared neural code for visually presented actions and object events throughout the action observation network. We argue that this neural representation captures the structure and physics of events regardless of animacy. We find that lateral occipitotemporal cortex encodes information about events that is also invariant to stimulus modality. Our results shed light onto the representational profiles of posterior temporal and frontoparietal cortices, and their roles in encoding event information.

Every day, we experience the world dynamically, not just as a set of static objects but as a series of changes, relations, and events. When a person moves their foot quickly and makes contact with a ball—we interpret it as A kicking B. When two billiard balls collide and one of them starts moving—we interpret it as A launching B into motion[1,2]. In order to support this understanding, our brains need to process and integrate various types of complex information ranging from the physical properties of agents, objects, and their movement to more abstract information such as an action's meaning or its goals. How are these various aspects of events encoded in the brain, and what are the roles of different regions in this process?

Functional neuroimaging has revealed a set of bilateral frontoparietal and posterior temporal regions that are consistently recruited when observing others' actions (e.g., someone jumping). These regions are collectively termed as the action observation network (AON), with its frontoparietal component also called the mirror neuron system[3–6]. With a particular emphasis on lateral occipitotemporal cortex (LOTC), inferior parietal lobe (IPL), and ventral premotor cortex (PMv), these regions are thought to play complementary roles in encoding information about observed actions[7–11].

However, not only humans and other animate entities but also inanimate objects can participate in events with rich meaning and

structure (object events, e.g., a ball bouncing). Furthermore, despite differences between actions and object events on several dimensions (e.g., biological motion, goals), a formal or linguistic description of events that is invariant to animacy can characterize both (e.g., A makes contact with B). Since previous work mainly focused on the neural representation of actions without a systematic comparison to object events, a neural representation that can capture both event types has not been identified. In this study, we searched for a shared neural code that can characterize both actions and object events by investigating the neural activity patterns associated with observing visually presented human actions (e.g., a boy jumping over a box) with structurally similar motion events of objects (e.g., a ball bouncing over a box).

Identifying a neural representation that can capture both actions and object events requires a systematic investigation of the two event types within a unified framework. Previously, in rare cases where actions and object events were investigated together, they tended to have highly distinct structural and perceptual properties (e.g., point light displays of bodies vs. tool motion[12], see ref. 13 for a review). Furthermore, object events were mostly used as baseline while testing the boundaries of categorical specificity for event components unique to animate entities (e.g., intentions, agency, sociality[14–16]). As powerful as such designs are to study how the unique properties of actions are

[1]Department of Psychology, Harvard University, Cambridge, MA, USA. [2]Center for Mind/Brain Sciences – CIMeC, University of Trento, Rovereto, Italy. ✉e-mail: sakbiyik@fas.harvard.edu

represented in the brain, they do not lend themselves to the investigation of a neural representation of events that is invariant to animacy.

There is a long tradition of research on structured event representations (i.e., event models) that enable predictive processing of complex naturalistic stimuli (see refs. 17–19 for reviews, see ref. 20 for a recent computational model). This work revealed neural representations of events that are shared across perception, memory, and language[21–24], which presumably could capture the shared aspects of actions and object events as well. However, previous work in this domain mostly focused on actions of humans, especially in the context of how the brain segments ongoing human activity into meaningful elements. Thus, a general neural representation of events that can capture both actions and object events has not been addressed explicitly. Finally, previous literature on action recognition mostly

compared the strength of neural responses to animate actions versus simple dynamic stimuli (i.e., univariate analysis). Such comparisons help reveal anatomical overlaps or discrepancies in neural responses, but they do not provide direct information regarding the representational content of different brain regions.

In the current study, we investigated where in the brain information about events is captured as a function of, or invariant to, the animacy of the subject of the event. To address this aim, we used classification techniques that approach neural responses as patterns of activity (e.g., multivariate pattern analysis and cross-decoding, see Fig. 1B). We used structurally similar events for actions of humans and motion events of objects, and classifiers were trained and tested to distinguish these events based on their respective neural activity patterns (e.g., walk-jump-kick for human actions and roll-bounce-hit for

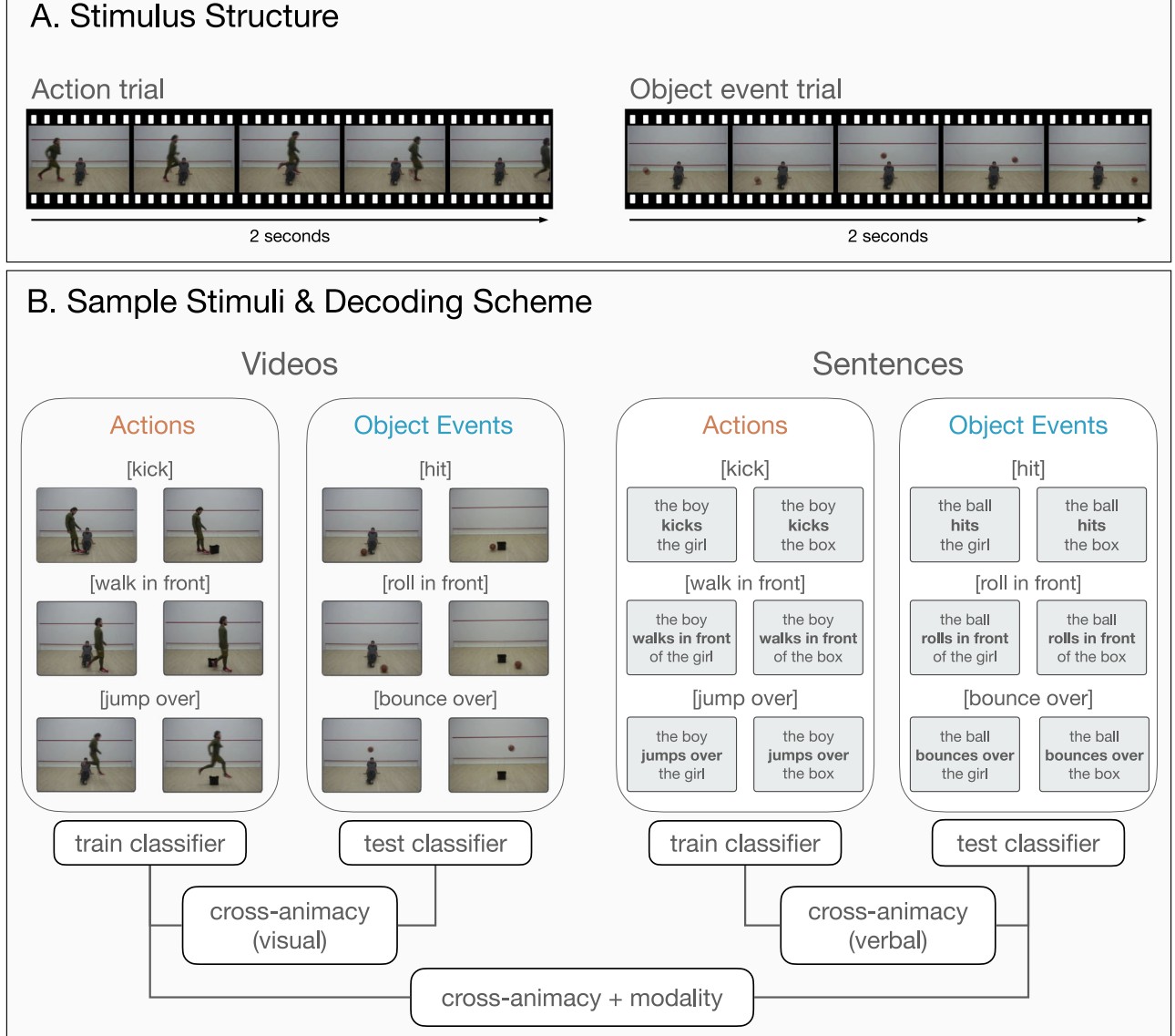

Fig. 1 | Sample stimuli and experimental design. A Example trials for action and object event stimuli in the video session. For both event types, 2-sec videos were used depicting the movements of a human or a ball. B Experimental design and classification scheme. For both actions and object events, three motion trajectories were used in relation to an animate or inanimate patient (i.e., kick/hit, jump over/bounce over, walk in front/roll in front of an object/person). To test for generalization across animacy within a modality, a classifier was trained on neural activity patterns associated with events from one category (e.g., action videos) and tested on neural activity patterns derived from events of the other category (e.g., object event videos). Training and testing were done in both directions and resulting accuracy maps were averaged. To test for generalization across both animacy and modality, a classifier was trained on events from one category in one modality (e.g., action videos) and tested on the other category in the other modality (e.g., object event sentences). Training and testing were done across combinations of modality and animacy conditions and the resulting accuracies were averaged across training and testing directions.

object events). To control for differences between observed actions and object events that do not pertain to the events themselves, we generated different exemplars for our unique event conditions by displaying them across different viewpoints, subjects, animate/inanimate passive patients, and moving directions. This approach also allowed us to reveal neural sensitivity to actions or object events in a way that is not confounded by the presence of humans in the scene. Finally, by using complementary evidence from verbal depictions of events, we also searched for a common neural representation of events that is not tied to the visual modality.

Here, we show that posterior temporal and frontoparietal regions previously linked to action recognition represent information about visually presented actions and object events in a similar way. Subregions in posterior superior/middle temporal sulcus and superior parietal lobes are more sensitive to agent-specific event aspects. Through cross-decoding, we also find a shared neural code for observed actions and object events throughout the action observation network, and a representation of events in LOTC that is also invariant to stimulus modality. Our results shed light onto the representational profiles of temporal and frontoparietal cortices and their roles in encoding event information.

## Results
### Procedure
To search for brain regions that host a shared neural code for observed actions of humans (actions) and motion events of objects (object events), we scanned participants ($n = 25$) while they viewed video clips of humans and balls moving in structurally similar ways (e.g., a person jumps over a box or a ball bounces over a box, see Fig. 1A, B for sample stimuli, see Supplementary Fig. 1 for univariate neural responses to observed actions and object events). To search for brain regions that host a shared neural code for actions and object events that is also invariant to stimulus modality, we collected complementary data from a sentence comprehension experiment. In this session, the same participants read Subject-Verb-Object sentences describing the events that are shown in the video session (e.g., the sentences 'The boy jumps over the box' or 'The ball bounces over the box', see Fig. 1B for sample stimuli, see Supplementary Fig. 2 for univariate neural responses to sentences describing actions and object events).

### Overlapping representations of actions and object events
First, we aimed to identify whether information about actions and object events is encoded in overlapping or distinct brain regions. To this end, we conducted two whole-brain multivoxel pattern analyses (MVPA): within-actions and within-object-events. For within-actions decoding, we trained and tested a classifier with neural activity patterns associated with actions (i.e., kick/jump/walk). For within-object-events decoding, we trained and tested a classifier with neural activity patterns associated with object events (i.e., hit/bounce/roll). We created different exemplars for each motion trajectory by using different viewpoints, subjects, patients, and moving directions to make sure that the decoding of events does not rely purely on low-level visual features (see Methods for more detail).

In line with previous findings[25–27], actions were decoded in extended networks spanning occipital, posterior temporal, frontal, and parietal cortices (Fig. 2A). Strikingly, object events were also decoded in closely overlapping brain regions (Fig. 2B). To obtain a better understanding of action and object event decoding across specific regions, we extracted classification accuracies from independently defined regions of interest (ROIs) in each hemisphere. We primarily focused on regions of the action observation network that are most strongly and consistently recruited during action observation tasks: lateral occipitotemporal cortex (LOTC), ventral premotor cortex (PMv), and inferior parietal lobe (IPL)[3,5]. To provide a more fine-grained picture of how actions and object events are

represented in other areas that are also linked to action observation, we also report ROI results from the superior parietal lobe (SPL) and posterior superior temporal sulcus (pSTS) (see Methods for more details on ROI selection).

Consistent with the whole-brain results (Fig. 2A, B), all ROIs in both left and right hemispheres showed above-chance decoding of the three events for both actions and object events (Fig. 2D, one-tailed $t$ tests against chance-level 33.33%, all $p$s < 0.001, FDR-corrected). Overall, these analyses revealed that overlapping brain regions of the so-called action observation network encode information about both actions and object events, in a remarkably similar way.

### Distinct representations of actions and object events
Structurally similar actions and object events can be defined by common physics and kinematics. However, animate actions are interpreted as not mere movements of a physical entity, but rather, intentional actions of a sentient being. That is, actions carry additional information specific to animate entities (e.g., biological motion, intentions, goals) that are not present in object events. To investigate where in the brain information specific to animate actions is encoded, we compared the outputs of within-actions and within-object-events decoding. The logic behind this analysis is that if a region encodes additional information about agent-specific event features that cannot be captured by an inanimate object's movements, it will show better decoding of actions compared to object events.

A two-tailed whole-brain paired $t$ test revealed a cluster in the right posterior superior to middle temporal sulcus (pSTS) and temporoparietal junction (TPJ) that could better distinguish actions compared to object events. Although additional clusters in bilateral superior parietal lobes (SPL) and ventral temporal cortices showed significantly higher decoding for actions than object events ($p$s < 0.005), these clusters did not survive correction for multiple comparisons in the whole brain (Fig. 2C). For a more fine-grained examination of the differences between the neural representation of actions and object events, we again turned to the ROI analysis.

To investigate if either of the two hemispheres is more sensitive to information about actions and object events, we fitted a linear mixed effect model testing the interaction of event type and hemisphere across all ROIs. The hemisphere by event type interaction was significant ($\chi^2[1] = 4.80$, $p = 0.028$, $\Delta AIC = 2.80$) and the difference in decoding accuracy between actions and object events was stronger in the right hemisphere ($b = 4.72$, $t(472) = 4.48$, $p < 0.001$, $d = 0.57$, 95% CI [0.32 0.82]) compared to the left hemisphere ($b = 2.53$, $t(472) = 2.40$, $p = 0.017$, $d = 0.30$, 95% CI [0.05 0.55]).

To compare decoding accuracies for actions and object events across different brain regions within each hemisphere, we fitted linear mixed effect models testing the interaction of event type and ROI for each hemisphere. This ROI by event type interaction was significant both in the left hemisphere ($\chi^2[4] = 26.06$, $p < 0.001$, $\Delta AIC = 18.06$) and in the right hemisphere ($\chi^2[4] = 31.13$, $p < 0.001$, $\Delta AIC = 23.13$). In the left hemisphere, actions and object events were classified with comparable accuracy in ventral premotor cortex ($b = 0.94$, $t(216) = 0.52$, $p = 0.601$, $d = 0.15$, 95% CI [−0.41 0.71]) and inferior parietal lobe ($b = −2.27$, $t(216) = −1.27$, $p = 0.207$, $d = −0.36$, 95% CI [−0.92 0.20]). However, actions were decoded at a higher accuracy than object events in left LOTC ($b = 3.85$, $t(216) = 2.14$, $p = 0.034$, $d = 0.61$, 95% CI [0.05 1.17]), pSTS ($b = 5.57$, $t(216) = 3.10$, $p = 0.002$, $d = 0.88$, 95% CI 0.31 1.44]), and SPL ($b = 4.55$, $t(216) = 2.53$, $p = 0.012$, $d = 0.72$, 95% CI [0.15 1.28]). In the right hemisphere, actions and object events were classified with comparable accuracy in ventral premotor cortex ($b = 0.40$, $t(216) = 0.25$, $p = 0.804$, $d = 0.07$, 95% CI [−0.49 0.63]) and inferior parietal lobe ($b = 1.54$, $t(216) = 0.94$, $p = 0.346$, $d = 0.27$, 95% CI [−0.29 0.83]). However, actions were decoded at a higher accuracy than object events in right LOTC ($b = 7.23$, $t(216) = 4.45$, $p < 0.001$, $d = 1.26$, 95% CI [0.69 1.83]), pSTS ($b = 8.23$, $t(216) = 5.06$, $p < 0.001$, $d = 1.43$, 95%

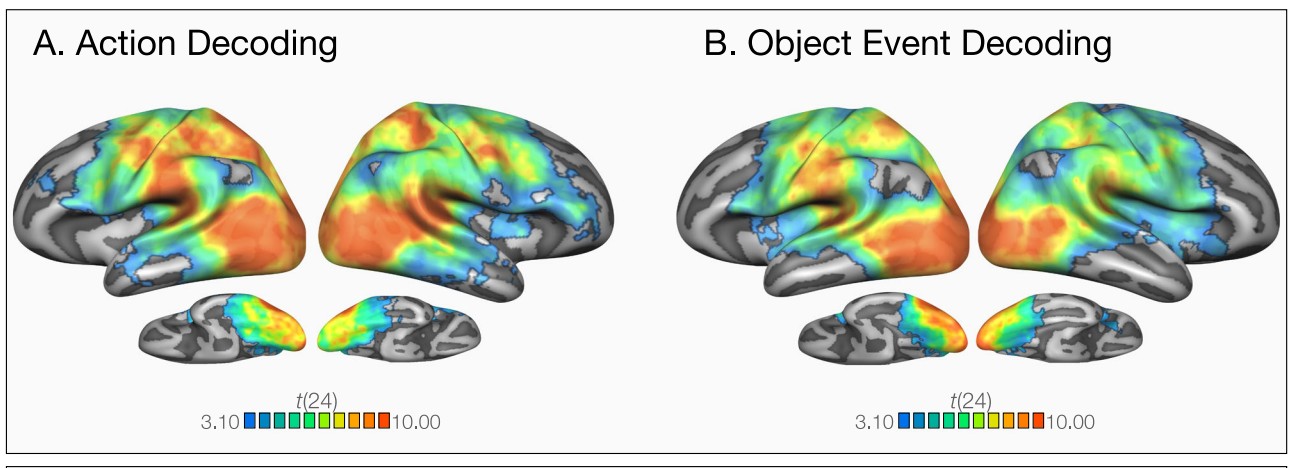

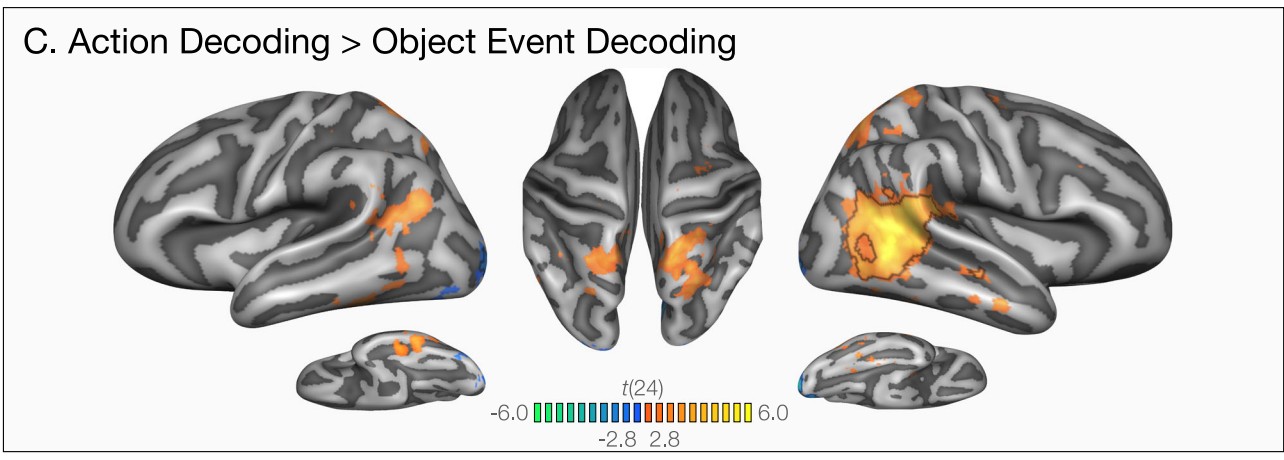

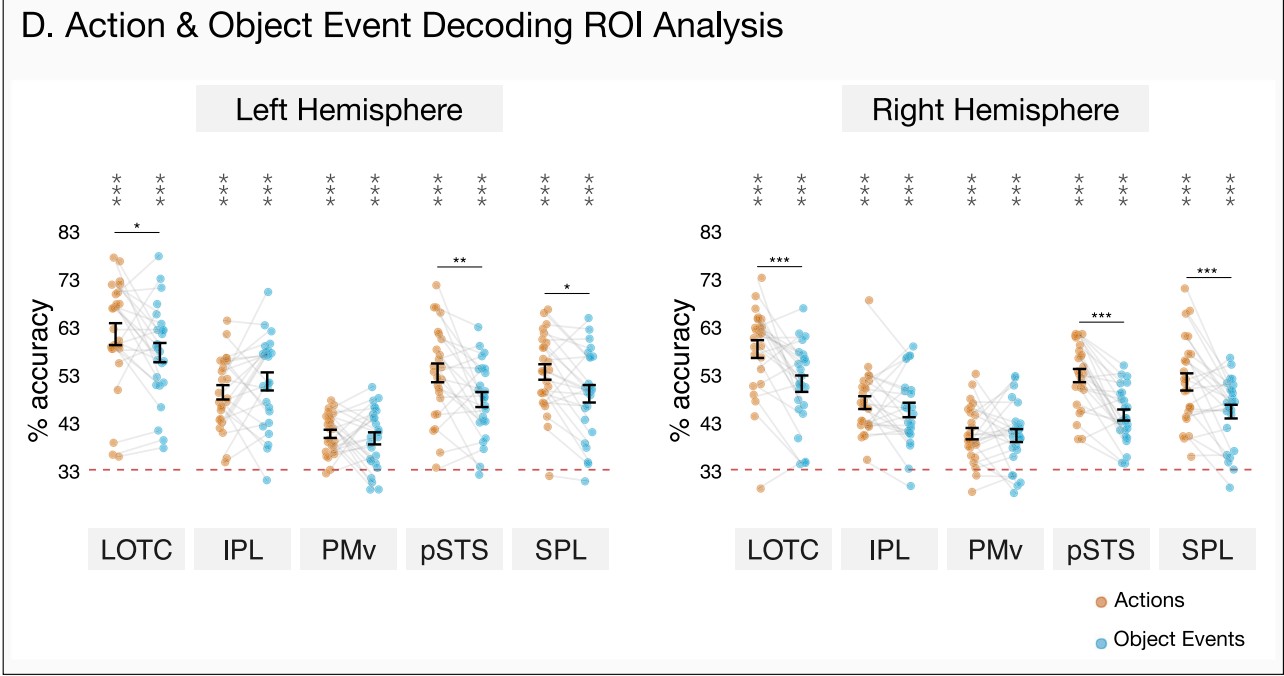

CI [0.86 2.01]), and SPL ($b = 6.20$, $t(216) = 3.81$, $p < 0.001$, $d = 1.08$, 95% CI [0.51 1.64]).

In sum, within-actions and within-object-events decoding analyses revealed overlapping brain regions that encode information about actions and object events, with minimal differences in decoding strength across regions within each hemisphere. In the ROI analysis, actions were classified at a higher accuracy than object events in LOTC,

pSTS, and SPL in both hemispheres. In the whole-brain analysis, this difference survived correction for multiple comparisons only in clusters around right LOTC and pSTS/TPJ (for a comparison of decoding of actions and object events separated by the animacy of the passive patient, see Supplementary Note 1, see Supplementary Fig. 3).

We found distinct neural representations of actions and object events in pSTS/TPJ and superior parietal lobes. Higher sensitivity to

**Fig. 2 | Within-condition decoding of actions and object events.** Results of whole-brain three-way decoding searchlight for **A** actions and **B** object events (one-tailed $t$ tests against chance-level 33.33%). The maps are thresholded by areas corrected for multiple comparisons using Monte Carlo Cluster based correction ($p_{initial} = 0.001$). **C** Two-tailed whole-brain $t$ test comparison of within-actions and within-object-events MVPA. Black outlines mark areas that survived Monte Carlo Cluster based correction ($p_{initial} = 0.001$). The map is thresholded at $p < 0.01$ to demonstrate significant differences that do not survive correction. **D** ROI decoding accuracies for actions (in orange) and object events (in blue) in lateral occipitotemporal cortex (LOTC), inferior parietal lobule (IPL), ventral premotor cortex (PMv), posterior superior temporal sulcus (pSTS), and superior parietal lobule (SPL). Error bars indicate standard error of the mean (SEM, $n = 25$), and asterisks indicate FDR-corrected effects of one-tailed $t$ tests for comparisons against chance-level (33.33%, *$p < 0.05$, **$p < 0.01$, ***$p < 0.001$). Individual participants are connected via light gray lines. FDR-corrected pairwise two-tailed tests of estimated marginal means showed better decoding of actions compared to object events in LOTC, pSTS, and SPL in both hemispheres (*$p < 0.05$, **$p < 0.01$, ***$p < 0.001$).

action information in right pSTS/TPJ is consistent with previous studies linking these regions to human-specific event information such as animacy and intentionality, social interactions, and biological motion[12,14-16,28-33]. However, we would like to note that actions and object events used in the current study varied along many dimensions, and some of these dimensions were orthogonal to the animacy of the actor. For instance, the movements of the human actors were richer in terms of visual information: the bodies had moving parts while objects did not. If a region is sensitive to variation in such differences in movement kinematics, changes in decoding strength across actions and object events could reflect encoding of such information.

## A shared neural code for actions and object events

Our within-actions and within-object-events decoding analyses revealed overlapping neural representations for actions and object events in regions linked to action recognition. However, overlap of decoding does not guarantee activation of shared neural representations[34,35]. That is, the overlap of decoding between actions and object events might have stemmed from the activation of spatially overlapping but functionally distinct neural populations.

To identify brain regions that encode a shared representation of events independent of whether the event is associated with the movements of an object or a person, we conducted cross-decoding MVPA. For this analysis, we trained a classifier to distinguish neural activity patterns associated with observed actions and tested its accuracy on observed object events, and vice versa (see Fig. 1B). By training a classifier to discriminate actions and testing the same classifier on its accuracy to discriminate the corresponding object events, this approach can identify spatially corresponding activity patterns of the two event types. Success in this generalization points toward event representations that are commonly defined by both actions and object events.

In the whole-brain, the cross-decoding analysis revealed robust generalization across observed actions and object events in both frontoparietal and posterior temporal cortices, as well as throughout the occipital cortex (see Fig. 3A). Consistent with the whole-brain results, all ROIs in both left and right hemispheres showed above-chance decoding of the three events by generalizing across animacy (see Fig. 3B, all $p$s < 0.001, FDR-corrected).

Comparing the ROIs across the two hemispheres, we observed some hemispheric differences in the strength of cross-animacy generalization. There was an ROI by hemisphere interaction ($\chi^2[4] = 15.26$, $p = 0.004$, $\Delta$AIC = 7.25): cross-animacy generalization was stronger in the left hemisphere compared to the right hemisphere for all ROIs (LOTC: $b = 4.57$, $t(216) = 3.62$, $p < 0.001$, $d = 1.02$, 95% CI [0.46 1.59]; IPL: $b = 5.92$, $t(216) = 4.70$, $p < 0.001$, $d = 1.33$, 95% CI [0.76 1.90]; pSTS: $b = 3.60$, $t(216) = 2.85$, $p = 0.005$, $d = 0.81$, 95% CI [0.25 1.37]; SPL: $b = 2.95$, $t(216) = 2.34$, $p = 0.020$, $d = 0.66$, 95% CI [0.10 1.22]; PMv: $b = 2.66$, $t(216) = 2.11$, $p = 0.036$, $d = 0.60$, 95% CI [0.04 1.16]) with the difference being more salient for LOTC and IPL. Overall, decoding of the events regardless of animacy was stronger in all frontoparietal and posterior temporal ROIs compared to their right hemisphere homologs. Stronger cross-animacy generalization in the left hemisphere might imply distinct roles of left and right hemispheres in encoding general versus agent-specific event aspects, respectively.

Looking at the ROIs within each hemisphere, highest decoding accuracies were observed for LOTC in both hemispheres (Left: LOTC–IPL: $b = 5.01$, $t(216) = 3.97$, $p < 0.001$, $d = 1.12$, 95% CI [0.56 1.69]; LOTC–PMv: $b = 12.88$, $t(216) = 10.21$, $p < 0.001$, $d = 2.89$, 95% CI [2.27 3.51]; LOTC–pSTS: $b = 5.53$, $t(216) = 4.39$, $p < 0.001$, $d = 1.24$, 95% CI [0.67 1.81]; LOTC–SPL: $b = 4.44$, $t(216) = 3.52$, $p < 0.001$, $d = 1.00$, 95% CI [0.43 1.56]; Right: LOTC–IPL: $b = 6.37$, $t(216) = 5.05$, $p < 0.001$, $d = 1.43$, 95% CI [.86 2.00]; LOTC–PMv: $b = 10.97$, $t(216) = 8.70$, $p < 0.001$, $d = 2.46$, 95% CI [1.86 3.06]; LOTC–pSTS: $b = 4.57$, $t(216) = 3.62$, $p < 0.001$, $d = 1.02$, 95% CI [.46 1.59]; LOTC–SPL: $b = 2.83$, $t(216) = 2.24$, $p = 0.033$, $d = 0.63$, 95% CI [0.07 1.19]). Despite these differences in decoding strength, all investigated regions of the action observation network were robust to generalization across actions and object events.

Control analyses revealed that generalization across actions and object events in posterior temporal and frontoparietal brain regions linked to action recognition persisted even when said actions did not include any objects and said object events did not include any humans (e.g., train on events that only had objects "ball bounces over the box" and test on events that only had humans "woman jumps over the boy", see Supplementary Fig. 4a). This suggests that the cross-decoding results cannot be explained away by the relevance of some object events for animate entities in the scene since cross-decoding persists even when the object events do not pertain to an animate being. Furthermore, by investigating cross-decoding accuracies for pairs of motion events (i.e., kick-hit VS jump-bounce; kick-hit VS walk-roll; jump-bounce VS walk-roll), we also found that cross-animacy generalization did not rely on one peculiar event being different from the rest. Cross-animacy decoding for pairs of motion events revealed qualitatively similar patterns of decoding across the core regions of the action observation network (see Supplementary Fig. 4b).

What shared aspects of actions and object events are captured by cross-animacy decoding? Various shared properties of actions and object events ranging from low-level visual and motion cues to higher-level spatiotemporal relations or semantic properties could all contribute to successful cross-animacy generalization of observed events. We created different event exemplars for each category so that decoding does not purely rely on low-level visual properties but naturally, there was still some variation in low-level visual features across our event categories (e.g., bounce/jump events involve horizontal movement while others did not, hit/kick events happened in one half of the scene while the others spanned the whole scene). Cross-animacy generalization in early visual processing regions (see Fig. 3A) is consistent with the possibility that encoding of such features can contribute to cross-animacy generalization. However, this does not necessarily mean that cross-animacy generalization in regions of the so-called action observation network purely stemmed from shared low-level visual properties. Decoding accuracies in most regions of the action observation network (e.g., LOTC, IPL, pSTS, SPL) were stronger than in early visual processing regions. Thus, we think it is unlikely that lower-level visual features are the main information driving cross-animacy generalization in the action observation network.

To investigate more directly what shared aspects contribute to generalization across animacy, we conducted an exploratory representational similarity analysis (RSA)[36]. This analysis revealed that

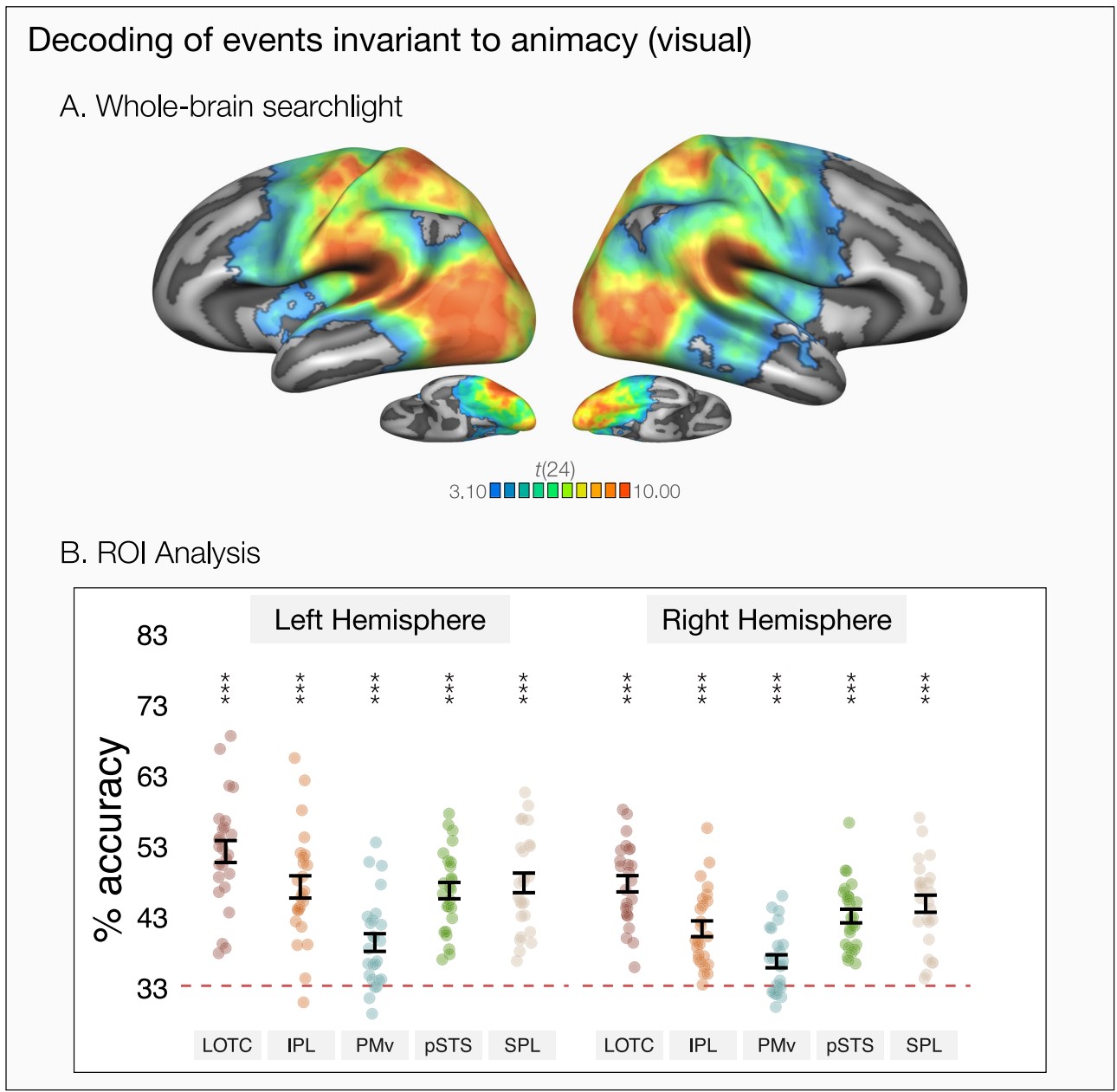

**Fig. 3 | Decoding of observed events by generalizing across animacy. A** Results of whole-brain three-way decoding searchlight by generalizing across actions and object events (one-tailed *t* test against chance-level 33.33%). The map is thresholded by areas corrected for multiple comparisons using Monte Carlo Cluster based correction ($p_{initial}$ = 0.001). **B** ROI decoding accuracies for cross-animacy generalization of video stimuli in lateral occipitotemporal cortex (LOTC), inferior parietal lobule (IPL), ventral premotor cortex (PMv), posterior superior temporal sulcus (pSTS), and superior parietal lobule (SPL). Error bars indicate standard error of the mean (SEM, *n* = 25), asterisks indicate FDR-corrected effects of one-tailed *t* tests for comparisons against chance-level (33.33%, * *p* < 0.05, ** *p* < 0.01, *** *p* < 0.001).

activity patterns throughout the action observation network, particularly in left anterior IPL, can capture inter-object relations such as making contact (see Supplementary Note 2, Supplementary Fig. 5a), whereas motion path (i.e., horizontal versus vertical movement) was captured only in dorsal occipital and medial premotor cortices (see Supplementary Note 2, Supplementary Fig. 5b). These preliminary analyses suggest that different regions of the action observation network may encode events at a level specifying inter-object relations.

**Generalization of actions and object events across modality**
To test if any of the regions that we identified in cross-animacy decoding in the video session encode event components that go beyond shared visual properties of observed events, we first replicated

our cross-animacy decoding analysis with sentence stimuli. That is, we trained a classifier to decode the three events on neural patterns from action sentences (e.g., the boy kicks the box) and tested it on neural patterns from object event sentences (e.g., the ball hits the box), and vice versa. Overall, classification accuracies tended to be lower for the sentence session compared to the video session. Thus, even though we report whole-brain results, we primarily focus on ROI analyses (i.e., left hemisphere action observation ROIs: LOTC, IPL, PMv, pSTS, SPL).

In the whole-brain analysis corrected for multiple comparisons, clusters in bilateral lateral and ventral occipitotemporal cortices, bilateral posterior parietal lobes, left inferior parietal lobe and left ventral premotor cortex showed above-chance cross-animacy decoding of sentences (see Fig. 4A). In the ROI analysis, all left hemisphere

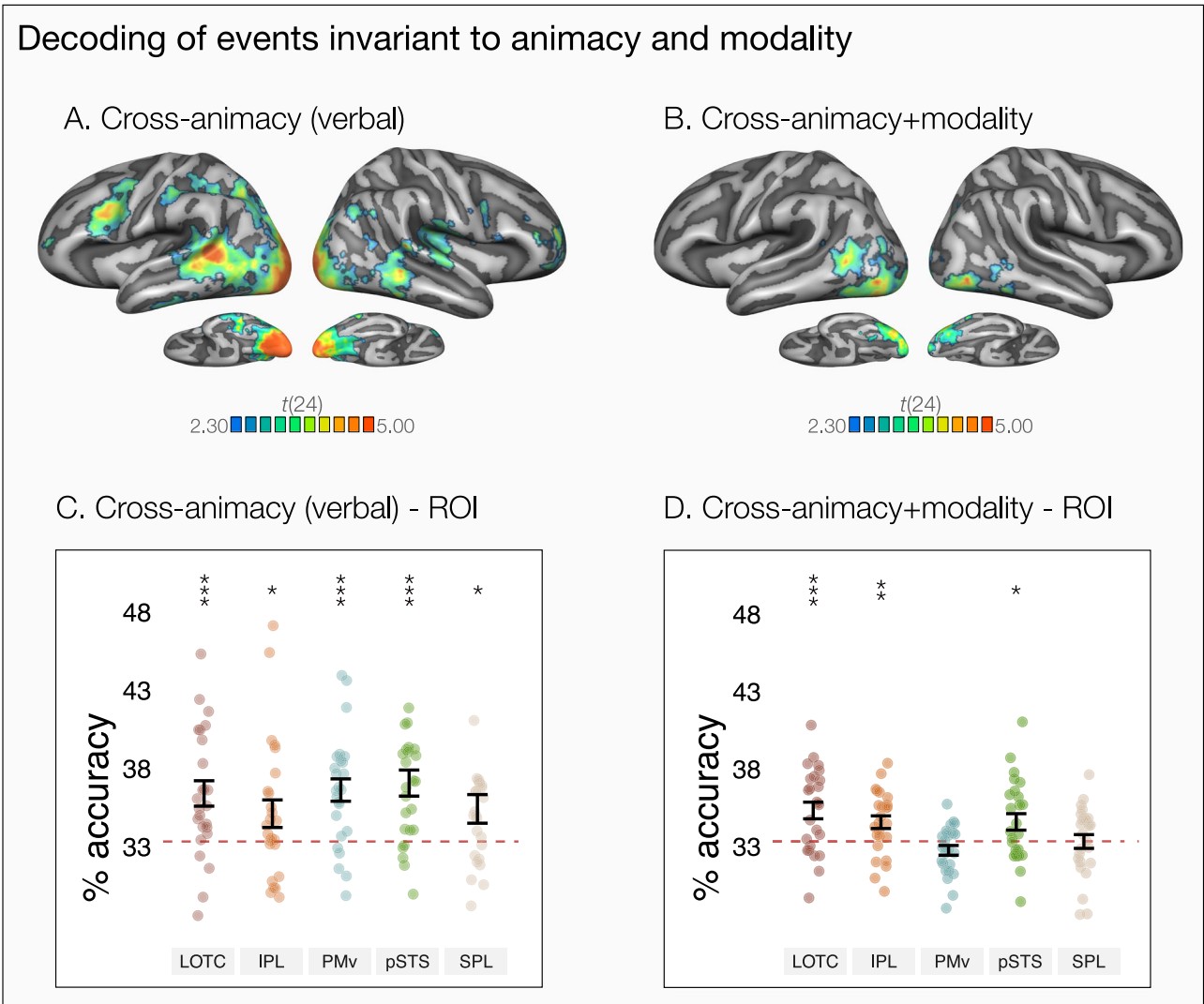

**Fig. 4 | Decoding of events by generalizing across animacy and modality.**
**A** Whole-brain three-way decoding of events by generalizing across animacy for
sentence stimuli (one-tailed $t$ test against chance-level 33.33%). **B** Whole-brain
three-way decoding of events by generalizing across both animacy and modality
(one-tailed $t$ test against chance-level 33.33%). The maps show areas corrected for
multiple comparisons using Monte Carlo Cluster based correction (outlined in
black, $p_{initial} = 0.005$). ROI decoding accuracies for cross-animacy generalization of

**C** sentence stimuli and **D** cross-animacy + modality generalization in in lateral
occipitotemporal cortex (LOTC), inferior parietal lobule (IPL), ventral premotor
cortex (PMv), posterior superior temporal sulcus (pSTS), and superior parietal
lobule (SPL). Error bars indicate (SEM, $n = 25$) and asterisks indicate FDR-corrected
effects of one-tailed $t$ tests for comparisons against chance-level (33.33%, *$p < 0.05$,
**$p < 0.01$, ***$p < 0.001$).

ROIs showed above-chance decoding, with minimal non-significant
differences between them (see Fig. 4C). In sum, left hemisphere
regions of the action observation network was able to discriminate
events described via verbal stimuli by generalizing across actions and
object events (for decoding of action and object event sentences
without generalization across animacy, see Supplementary Note 3,
Supplementary Fig. 6a, b).

Note that even though cross-animacy generalization in the sen-
tence session might reflect encoding of higher-level semantic infor-
mation, successful cross-decoding in the sentence session might have
been driven by certain stimulus characteristics such as sentence length
or presence/absence of prepositions that do not pertain to the
meanings of the event per se (see Supplementary Files for the full set of
sentence stimuli). For instance, 'kick-hit' sentences did not contain
prepositions and were shorter than 'jump-bounce' and 'walk-roll' sen-
tences, leading to differences in sentence length, and thus, perceptual
differences in the visually presented sentences. To test more explicitly
where in the brain events are represented at a more abstract level that

not only generalizes across actions and object events but also stimulus
modalities, we conducted cross-animacy + cross-modality decoding.
For this analysis, we trained a classifier on videos of one condition and
tested it on the sentences of the other condition (e.g., train on video
showing [The boy jumps over the box] and test on sentence 'The ball
bounces over the box', see Fig. 1B). Success in this generalization
points toward abstract event representations that are commonly
defined by both actions and object events, and visual and verbal sti-
muli (for cross-modality generalization within event types, see Sup-
plementary Fig. 6c, d).

In the whole-brain analysis, clusters in left LOTC extending to
pSTS, as well as some clusters in bilateral ventral occipitotemporal and
visual association cortices showed above-chance cross-animacy +
modality decoding that survived correction for multiple comparisons
(see Fig. 4B). The ROI analysis revealed that cross-animacy + modality
decoding was significantly above-chance in left LOTC, pSTS, and IPL,
but not in PMv or SPL (see Fig. 4D). Comparing the decoding accura-
cies across the different ROIs, we observed differences in decoding

strength ($\chi^2[4] = 30.68$, $p < 0.001$). While left LOTC showed the highest decoding accuracy and was significantly higher than PMv ($b = 2.58$, $t(96) = 4.82$, $p < 0.001$, $d = 1.36$, 95% CI [0.77 1.95]) and SPL ($b = 2.00$, $t(96) = 3.75$, $p = 0.002$, $d = 1.06$, 95% CI [0.48 1.64]), no significant differences were observed between LOTC and IPL ($b = 0.76$, $t(96) = 1.42$, $p = 0.21$, $d = 0.40$, 95% CI [−0.16 0.96]) or pSTS ($b = 0.74$, $t(96) = 1.38$, $p = 0.21$, $d = 0.39$, 95% CI [−0.17 0.95]).

In the so-called action observation network, only left LOTC/pSTS showed significant cross-animacy + modality generalization that survived correction for multiple comparisons in the whole brain. Is this region causally involved in understanding events both through visual and verbal formats? If cross-animacy + modality generalization in LOTC is due to verbalization (in the video session) or visual imagery (in the sentence session), this might not be the case. For instance, if generalization across observed videos and sentences is due to imagery in the sentence session, this would not guarantee that neural activity during sentence processing is essential for understanding events in that modality. Since fMRI can only provide correlational evidence, evidence from individuals with brain damage or neuromodulation studies can be used to address the critical role of this region in understanding events through visual and verbal modalities. Still, our study allowed us to address the extent of imagery or verbalization effects as we balanced the order of video and sentence sessions across participants. If across modality generalization is due to imagery, we would expect stronger imagery for people who received the videos first and sentences second, and stronger verbalization for people who received sentences first and videos second. The contrast of the decoding maps of the two groups, however, revealed no significant differences indicative of an effect of imagery or verbalization (see Supplementary Fig. 7). Thus, we found no support for the hypothesis that visual imagery or verbalization can fully account for the observed generalization across animacy and modality.

Successful cross-animacy + modality generalization in LOTC/pSTS is consistent with recent evidence for cross-modal event representations in this region[35]. Notably, we found cross-animacy + modality decoding not only in left LOTC, but also in posterior and ventral occipital areas in both hemispheres (see Fig. 4B). This finding is consistent with previous evidence showing cross-modal decoding of object categories[37–40] or cross-task generalization of action similarity[41] in posterior occipital regions. However, since we cannot rule out potential low-level visual similarities across videos and sentences (e.g., visual extent of stimuli, eye movement), cross-animacy + modality decoding in posterior occipital regions must be interpreted with caution.

In the ROI analysis, left IPL also showed cross-animacy + modality generalization, which is consistent with previous claims that parts of IPL may contain conceptual representations. For instance, previous studies revealed representations of actions in IPL that generalize across specific instances of observed actions, viewpoints, and visual and motor modalities[26,27,42,43]. Notably, a recent study by Wurm & Caramazza[35] found generalization across observed videos and sentences only in left LOTC, but not in IPL. This discrepancy between Wurm & Caramazza[35] and the current study might be attributed to the kinds of actions that were tested (i.e., hand actions that were defined by manner of movement vs. body actions or object events that were defined by path of motion, respectively). Overall, more work is needed to address the differential roles of LOTC and IPL in encoding event information within and across stimulus formats.

## Discussion

Using multivariate pattern analysis techniques, we show that overlapping brain regions in posterior temporal and frontoparietal cortices encode information about observed actions and object events in a similar way. Using cross-decoding, we also provide robust evidence for a neural representation of observed events that is invariant to animacy.

Subregions in lateral occipitotemporal cortex represent events by generalizing across both animacy (i.e., humans and objects) and stimulus modality (i.e., visual and verbal stimuli). Yet, right posterior superior/middle temporal sulcus and temporoparietal junction as well as bilateral parietal lobes are particularly sensitive to information about observed human actions compared to object events. Overall, while conceptual event representations converged in left posterior temporal cortex, encoding of human-specific event information was primarily right-lateralized.

Premotor cortex and inferior parietal lobe have long been associated with encoding of action knowledge, particularly in relation to their motor functions[44–46]. However, in the current study, object events devoid of sensorimotor features of bodily actions were decoded as robustly as actions in both premotor cortex and IPL. Furthermore, both regions could represent event information by generalizing across humans and inanimate objects. What might be the driving force behind this general event representation in frontoparietal cortices? Previous work showed that frontoparietal regions are engaged in tasks that require physical inference[47,48] and predicting future steps of an event[49]. Based on the impairment patterns of apraxia patients who struggle with object use, it has also been proposed that the IPL is implicated in representing spatial relations between objects and mechanical reasoning[50]. Premotor cortex and IPL have also been linked to visual perception of causal interactions between objects[51,52]. Combined with our finding of a common neural representation for actions and object events in frontoparietal cortices, we think that these regions might be more properly construed as representing the physics and kinematics of events even if they lack any motor-relevant properties.

There has been growing evidence for encoding of higher-level action knowledge in LOTC (see refs. 11,53 for reviews). For instance, this region encodes information about actions by generalizing across their concrete instantiations (i.e., opening a box and opening a bottle are encoded similarly)[42] or stimulus modalities (e.g., visual and verbal stimuli)[35]. Our findings expand this previous work by identifying a neural representation of events in the LOTC that is invariant to both animacy of the entities that are involved and stimulus modality. This functional profile of LOTC points to the possible role of this region in encoding higher-level semantic information about events. For instance, both actions and object events can be described through event primitives that are invariant to the animacy of the entities that are involved (e.g., make contact with, cause)[54–56]. Lateral occipitotemporal cortex seems to encode events at a sufficiently abstract level that can capture such semantic relations between agents, objects, and their environment.

Actions share many features with object events, but by being events that include intentional animate entities, they also have some unique properties. How are these unique aspects of actions represented relative to those properties shared across actions and object events? In the current study, posterior superior temporal sulcus, temporoparietal junction, and superior parietal lobes showed better decoding of actions compared to object events. Higher sensitivity to action information in posterior superior temporal sulcus and temporoparietal junction, especially in the right hemisphere, is consistent with previous work linking these regions to human-specific event information such as animacy and intentionality[14,16,29,30], social interactions[15,31–33] and biological motion[12,28]. As for superior parietal lobe, considering its link to visuomotor coordination[57–59], recognizing actions compared to object events might have activated motor representations in this region. Furthermore, superior parietal lobe has been linked to attentional and visuospatial processing[60–62] and recognizing and predicting future steps of intentional animate acts might recruit special predictive and attentional processes.

Our findings suggest potential research directions that will shed light on how the brain represents dynamic information about agents

and objects. An avenue for future work is identifying which shared aspects of actions and object events are encoded by their common neural representation in frontoparietal and posterior temporal cortices. Candidate features include, but are not limited to, shared visual and kinematic information, physics of visual scenes, and high-level semantic information. Another alternative is that the propensity to attribute human-like characteristics to inanimate entities, anthropomorphizing, underlies the common neural representation of actions and object events that we have uncovered[1,2,16]. Spontaneous recruitment of regions involved in social cognition while reasoning about movements of simple visual shapes[63–65] supports this possibility. According to this alternative, the cross-animacy representations we uncovered would not necessarily reflect encoding of event components that are shared across actions and object events, but rather, encoding of human-like properties that are attributed to inanimate objects' movement. Although possible, we think that this is highly unlikely in the context of our experiment since we used filmed events. Thus, there was no question as to the inanimate nature of the objects in the object event condition, nor did they move in ways that might signal animacy or intentionality (e.g., autonomous motion that appears to be goal-directed). Still, addressing the contributions of anthropomorphic reasoning to the shared neural representations of actions and object events will be an important challenge for future work.

In addition, even though we think that common physics and spatiotemporal characteristics of events underlie the shared neural code we have uncovered, we would like to note that we have tested a small set of events that were primarily defined by their motion trajectories. This limits the conclusions we can derive regarding the underlying factors of cross-animacy generalization. Most events we perform and encounter in our daily life consists of small units organized in a temporal and spatial hierarchy. Whether the neural locus and degree of generalization across actions and object events will extend to these more complex scenarios remains to be seen. For instance, it has been shown that during passive viewing of daily activities or comprehension of narrative texts, posterior medial network regions (i.e., parahippocampal cortex, angular gyrus, medial prefrontal cortex, posterior cingulate, precuneus) and hippocampus are sensitive to information about event dynamics[66–69]. While these regions capture event dynamics at a longer temporal scale, regions that we focused here (e.g., posterior temporal cortex, inferior parietal lobe, and ventral premotor cortex) capture event boundaries at shorter temporal scales[70]. Even though we observed cross-animacy generalization in certain posterior medial regions (e.g., posterior medial parietal lobe, angular gyrus), we did not observe robust cross-animacy generalization in regions such as parahippocampal cortex or medial prefrontal cortex. Future studies can test events across different timescales and levels of complexity to address shared neural representations of actions and object events in different scenarios.

Finally, even though the physics and kinematics of events can commonly describe activities of both animate and inanimate entities, there are also key differences between them. For instance, agents can spontaneously change their path or speed, or fight against physical forces. However, the movements of objects are purely constrained by other agents or physical forces. Thus, predicting future steps of intentional animate acts might recruit special predictive and attentional processes. For a complete understanding of event recognition, more work is needed on how our brains integrate our knowledge of agents and their unique characteristics with that of the objects and the physical structure of the world. Future work can also delineate what aspects of actions are driving the differences between actions and object events in pSTS and SPL. Compared to object events, action stimuli in the current study involved biological motion and intentional movement and were richer in kinematic information. These regions might be sensitive to all or some of these features and discovering their

unique functional characteristics will provide significant avenues for future work. Furthermore, in posterior temporal cortices, event-general representations tended to be left-lateralized while encoding of human-specific information was right-lateralized. The underlying basis for this hemispheric lateralization remains to be worked out.

Overall, our results provide insights on how information about dynamic events is encoded in the brain across stimulus modalities and types of entities that are involved. Even though different neural systems might be recruited while processing information about actions and object events (as evidenced by differences in pSTS, TPJ, and SPL), regions that encode information about observed actions carry a shared neural code that can represent events more broadly. By providing clear evidence for a shared neural code for actions and object events, our study highlights the broader role of regions classically associated with action recognition in encoding the physics and kinematics of events, regardless of whether they involve people or objects. Our findings thus invite rethinking the common interpretation that neural responses to actions are naturally due to action-specific, or motor, aspects. This work could be a starting point for systematic research that will delineate how the brain encodes information about events ranging from low-level perceptual information about agents' and objects' movement to more abstract information such as an action's meaning or its goals.

## Methods

### Participants
Twenty-five right-handed native Italian speakers participated in a video (Experiment 1) and sentence session (Experiment 2, 15 male, Mage = 24.52, SDage = 4.80). The order of sessions was counterbalanced across participants (odd IDs: video-sentences, even IDs: sentences-videos). All participants had normal or corrected-to-normal vision and no history of neurological or psychiatric disease. The participants provided informed consent before participation and all procedures were approved by the Ethics Committee for research involving human participants at the University of Trento, Italy. Participants were compensated with 30 euros.

No statistical test was used to predetermine sample size. We collected data from $n = 25$ participants who attended both the video and sentence sessions. Our previous work on human action observation showed that depending on region of interest, actions presented in videos can be decoded with sample sizes between $n = 5$ (left LOTC; with $d = 1.94$, alpha = 0.05, power = 0.95) and $n = 14$ (left PMC; with $d = 0.95$, alpha = 0.05, power = 0.95)[27]. Another recent study from our lab showed cross-decoding of human actions across verbal and visual stimuli with $n = 22$ participants[35]. Given these considerations, we think that our sample size of $n = 25$ is in the range of sample sizes conventionally used in the field and gives us sufficient power to identify where in the brain information about events are encoded.

### Stimuli
For both video and sentence sessions, there were 6 unique events of interest that varied along two main dimensions: the animacy of the moving entity (animate/inanimate) and the motion trajectory (e.g., jump-bounce, walk-roll, kick-hit). For the video session, 32 exemplars of these 6 unique events were used, resulting in 192 unique videos in total. In both actions and object events, there was a passive animate or inanimate entity in the scene (i.e., patient). These passive patients served as reference for the trajectory of the motion event (e.g., the ball rolled in front of a box/girl). Furthermore, we used both animate and inanimate entities as passive patients to make sure that any differences between actions or object events cannot be explained away by the mere presence or absence of humans in the scene. Sample stimuli are presented in Fig. 1A, B and Supplementary Figs. 3–4. Individuals shown in these figures provided informed consent for the publication of these images in the context of this article.

We created different exemplars for each event to make sure that encoding of events are not trained purely on perceptual features. This perceptual variance was introduced via two different perspectives, two subjects, animate/inanimate patients of two exemplars each, and two moving directions for each unique event (32 exemplars, see Supplementary Files for event exemplars). The videos were shown at least twice over the course of the experiment. All videos were comparable in terms of the timing and unfolding of events (see Fig. 1A). They all started with one passive patient in the scene, followed by a motion event of a human or a ball. Videos were presented in color, had a length of 2 seconds (30 frames per second), and a resolution of 274 by 367 pixels.

The sentence stimuli matched the video stimuli in terms of stimulus variance (32 sentences of 6 events: 192 unique sentences in total). All sentences had the structure subject-verb-object (e.g., Il ragazzo calcia la donna/The boy kicks the woman; Il pallone colpisce la donna/The ball hits the woman). Different verbs were used to describe the actions and object events (action verbs: calcia/kick, salta oltre/jump over, cammina davanti/walk in front; object event verbs: colpisce/hit, rimbalza oltre/bounce over, rotola davanti/roll in front). To create 32 exemplars per event type, we crossed these verb phrases with four subjects (agents: lei/she, lui/he, la ragazza/the girl, il ragazzo/the boy; objects: la palla da pallavolo/the volleyball, il pallone/the football, la palla da basket/the basketball, la palla di gomma/the rubber ball) and eight patients (animate patients: l'amica/girlfriend, l'amico/boyfriend, l'uomo/the man, la donna/the woman; inanimate patients: lo sgabello/the stool, la sedia/the chair, la panca/the bench, il cassone/the box). All sentences were presented superimposed on light gray background (274 by 367 pixels) in three consecutive chunks (subject, verb phrase, object), with each chunk shown for 666 msec (2 s per sentence), using different font types (Arial, Comic Sans MS, Verdana, MV Boli, Times New Roman, Calibri Light) and font sizes (25–30) to increase the perceptual variance of the sentence stimuli (balanced across conditions within experimental runs). Since our analyses on the shared aspects of actions and object events relied on generalization across stimulus types, perceptual and syntactic differences between sentences, such as sentence length and presence/absence of prepositions, were ignored.

Stimuli were back-projected onto a screen (60 Hz frame rate, 800 × 600 pixels screen resolution) via a liquid crystal projector (OC EMP 7900, Epson Nagano, Japan). While in the scanner, participants viewed stimuli through a mirror mounted on the head coil. Stimulus presentation, response collection, and synchronization with the scanner were controlled with the MATLAB Psychtoolbox-3 for Windows.

## Task

Before fMRI, participants were instructed and trained for their respective first session (videos or sentences). The instructions and the practice of the second session were completed inside the scanner and after the first session. Participants were instructed to watch the videos (or read the sentences) and press a button with their right index finger on a response button box when they detected occasionally presented aberrant videos or sentences (catch trials, 14% of trials). The catch trials ensured that participants paid attention and understood the events. For videos, these aberrations were either conceptual or perceptual. In conceptual catch trials, the video depicted an incomplete event or an event that followed a different motion path than the three main events. In perceptual catch trials, a visual oddity was introduced to the video (e.g., freezing). For sentences, catch trials comprised grammatically incorrect or semantically odd versions of each of the six events (e.g., The boy walks in front for the woman, The ball bounces over the sun). The different types of catch trials were shuffled and randomly distributed across all runs. Responses made between the onset of the video and prior to the end of the 1 s fixation period following each trial were counted.

Participants detected catch trials with high accuracy in both video (Median = 0.90, $SD = 0.16$) and sentence sessions (Median = 0.86, $SD = 0.11$), with no difference in performance between them ($t(24) = −0.40$, $p = 0.691$, $d = −.08$, 95% CI [−.48 .32]). In both sessions, false alarm rates were low (video: Median = 0.005, SD = 0.017, sentence: Median = 0.005, SD = 0.019) indicating that participants paid attention to the task and did not confuse experimental trials with catch trials. Not only was the false alarm rate low, in both video and sentence sessions, participants were equally likely to make a false alarm for actions and object events (Video: $t(24) = 0.86$, $p = 0.396$, $d = 0.18$, 95% CI [−.23 .58]; Sentence: $t(24) = 1.86$, $p = 0.075$, $d = 0.38$, 95% CI [−.04 .79]). This suggests that the catch trial detection task imposed comparable demands for action and object event stimuli. Behavioral accuracies 2 SD below the average were used as a pre-established exclusion criterion. Based on this criterion, no data were excluded from the analyses.

## Experimental design

For both experiments, stimuli were presented in a mixed event-related design. The video and sentence experiments consisted of four and five functional scans, respectively. Each functional scan started with a 10 s fixation period and ended with a 16 s fixation period. Four blocks were presented per run, separated by 10 s fixation periods. Twenty-eight trials were shown per block. Each of the 6 unique events was presented four times per block, along with four catch trials. In every trial, videos or sentences (2 s) were followed by a 1 s fixation period. In the video session, for each of the 6 unique events, there were 64 trials in total (4 trials per block × 4 blocks per run × 4 runs per session). For sentences, for each of the 6 unique events, there were 80 trials in total (4 trials per block × 4 blocks per run × 5 runs per session. The order of conditions was counterbalanced within runs.

## Data acquisition

Neuroimaging data were acquired using a 3 T Siemens Prisma fMRI Scanner with a 32-channel phased-array head coil. T1-weighted structural images were obtained using a 3D MPRAGE sequence (176 sagittal slices; repetition time (TR) = 2530 ms; inversion time = 1020 ms; flip angle = 7 degrees; field of view (FoV) = 256 × 256 mm; 1 × 1 × 1 mm voxel resolution). Blood oxygenation level-dependent (BOLD) contrast functional images were obtained using a T2*-weighted gradient echo-planar imaging (EPI) sequence (TR = 1500 ms; echo time (TE) = 28 ms; inter slice time = 33 ms; flip angle = 70 degrees; FoV = 200 mm × 200 mm; matrix size = 66 × 66; 3 × 3 × 3 mm voxel resolution; 45 slices with 3-mm thickness and 0 mm gap).

## Preprocessing

We preprocessed and analyzed functional and anatomical data using BrainVoyager QX 2.8 (BrainInnovation), NeuroElf Toolboxes, and MATLAB (MathWorks) functions. The first four volumes of functional runs were removed to prevent T1 saturation. Preprocessing of functional data included slice time correction, three-dimensional motion correction (trilinear interpolation, the first volume of the first run of each participant was used as reference), linear trend removal, high-pass filtering (cutoff frequency of three cycles), and spatial smoothing (Gaussian kernel of 8 mm FWHM for univariate analyses and 3 mm FWHM for MVPA). Functional images were registered to high-resolution anatomical images (six parameters), and anatomical and functional data were normalized to Talairach space.

## Event classification

For each participant, session (i.e., video or sentence), and run, we computed a general linear model using design matrices containing 24 event predictors (separate predictors were fit for animate/inanimate patients per 6 unique events resulting in 12 conditions, and two predictors were fit for each of these conditions based on 4 trials selected

from the odd and 4 selected from the even trials of that condition within a run), the catch trials, and the 6 parameters resulting from 3D motion correction (x, y, z translation and rotation). Regressors were defined as boxcar functions convolved with a canonical double-gamma hemodynamic response function. Trials were modeled as epochs lasting from video or sentence onset to offset (2 s). The resulting reference time courses were used to fit the signal time courses of each voxel. In total, this procedure resulted in 16 beta maps per 6 unique events in the video, and 20 beta maps per 6 unique events in the sentence session.

We performed searchlight classification for each participant separately in volume space using searchlight spheres with a radius of 12 mm and a linear support vector machine (SVM) classifier as implemented by the CoSMoMVPA toolbox and LIBSVM[71,72]. Data were demeaned for each multivoxel beta pattern within a searchlight sphere by subtracting the mean beta of a sphere from the betas of the individual voxels. Demeaning was applied to make sure that classifiers do not distinguish actions based on global univariate differences across ROIs due to different processing demands (e.g., differences in sentence length, spurious visual differences across scenes).

To identify where in the brain information about actions and object events are encoded, we conducted within-actions or within-object-events MVPA. In within-actions MVPA, we decoded the three actions via leave-one-out cross-validation: we trained a classifier to discriminate the three actions (i.e., kick, jump, walk) by using betas from all runs except one and tested its accuracy at discriminating the three actions on the betas of the held-out run. This procedure was completed in iterations to make sure that all patterns were included for both training and testing, and the resulting classification accuracies were averaged. We then repeated the same procedure for object events.

To identify the shared neural representations of actions and events that generalize across animacy or modality, we completed cross-decoding. For cross-decoding, the classifier was trained on all betas from one condition (e.g., trained to distinguish kick/walk/jump in actions), and then tested on all betas from another condition (e.g., tested to distinguish hit/roll/bounce in object events). This was then repeated in the opposite direction, and classification accuracies were averaged. Note that for all decoding analyses, motion events with different subjects, animate or inanimate passive patients, viewpoints, and moving directions were specified as the same event. That is, we collapsed the events across these variations, meaning the classifier was trained to detect the event independent of variation in these factors. Overall, this approach ensured that the neural representations of actions and object events we identify are not tied to the number of entities in the scene, presence/absence of humans, presence/absence of interpretable movement, or fine-grained perceptual information. Controlling for such factors would be difficult in a standard univariate approach, emphasizing the strength of the decoding approach for the current study.

Individual accuracy maps were entered into one-tailed one-sample $t$ tests to identify voxels that showed above-chance classification. Statistical maps were thresholded using Monte Carlo Cluster correction for multiple comparisons (10000 simulations). For Monte Carlo Cluster correction, we used an initial threshold of $p = 0.001$ for within-video classifications (see Figs. 2 and 3). Due to decreased signal-to-noise ratio, we used an initial threshold of $p = 0.005$ for within-sentence and cross-modal classifications (see Fig. 4). We projected maps on a cortex-based aligned group surface for visualization using a transparency factor of 0.8 for visibility of gyri and sulci.

## ROI analysis
For a more fine-grained understanding of action and object event decoding across frontoparietal and posterior temporal cortices, we performed ROI analyses on areas that are commonly recruited during action observation. We selected the relevant coordinates based on a meta-analysis of action observation, which revealed increased activity

across a range of frontal, temporal, and parietal brain regions during action observation tasks[3]. Among these regions, we primarily focused on a bilateral network of three core regions of the so-called "action observation network"—the lateral occipitotemporal cortex (LOTC), the inferior parietal lobule (IPL), and the ventral premotor cortex (PMv) – that are most strongly and consistently recruited during action observation tasks (see refs. 3,5 for reviews). In addition to these three core regions of the AON, for a more fine-grained understanding of differences between actions and object events, and cross-animacy generalization in AON more broadly, we also report results from the superior parietal lobe and posterior superior temporal sulcus. Since the meta-analysis provided multiple ROIs for superior parietal lobe, for simplicity, we used the centroid of Brodmann areas 7 for left and right visuomotor SPL (see ref. 73). The MNI coordinates from the meta-analysis were converted to TAL coordinates using Yale Bioimage Suite[74,75], and all ROIs were created as spheres with a 12 mm radius around their respective coordinates (TAL coordinates: left LOTC [−45 −71 6], left IPL [−58 −23 34], left PMv [−48 8 29], left pSTS [−52 −49 11], left SPL [−18 −57 50]; right LOTC [52 −63 5], right IPL [44 −31 41], right PMv [50 12 27], right pSTS [54 −40 8], right SPL [24 −56 54]).

From each ROI, classification scheme (e.g., action decoding, object event decoding), and participant, we extracted decoding accuracies from the searchlight maps and averaged accuracies across voxels. FDR-corrected one-tailed $t$ tests were used to test above-chance classification for each ROI. To test for differences between actions and object events, we entered mean decoding accuracies into linear mixed effects models[76], and two-tailed post-hoc contrasts were implemented by estimated marginal means. To examine differences in classification accuracy, we fitted linear mixed effects models testing the interaction of event type, ROI, and if relevant, hemisphere, nested within subjects (i.e., subject ID was defined as a random effect). We assessed the residual normality and homoscedasticity assumptions for each model by inspecting residual plots. Upon inspecting the residual plots, we found no major deviations from normality or homoscedasticity, validating the use of these models. To test for interactions, say for the interaction of event type and ROI, we compared a model where ROI and event type were not allowed to interact with an expanded model where ROI and event type were allowed to interact (example R model syntax: Model 1: Classification Accuracy + Region + Event Type + (1|Subject ID); Model 2: Classification Accuracy ~ Region * Event Type + (1 | Subject ID)). We compared these models using a likelihood ratio test to examine if a model with an interaction term led to a better fit. Following the analyses of interaction, we conducted post-hoc contrasts to investigate the conditions driving the effects by using pairwise two-tailed estimated marginal means. Correction for multiple comparisons was conducted using the FDR method.

### Reporting summary
Further information on research design is available in the Nature Portfolio Reporting Summary linked to this article.

## Data availability
Raw and preprocessed functional neuroimaging data, design matrices, sample stimuli, and the Source data underlying Figs. 2–4 and Supplementary Figs. 3c, 6e, f are deposited at the Open Science Framework (https://osf.io/h4mtp/). The full set of stimuli are available from the corresponding author upon request.

## Code availability
Analysis code is available from the corresponding author upon request.

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

## Acknowledgements

This research was supported by the Caritro Foundation, Italy. We thank the members of the Cognitive Neuropsychology Lab at Harvard University and Cambridge Writing Group for their valuable feedback.

## Author contributions

S.K.-A., A.C., and M.F.W. developed the study design, S.K.-A. created the stimuli, M.F.W. collected the data, S.K.-A. and M.F.W. analyzed the data, and S.K.-A., A.C., and M.F.W. wrote the manuscript.

## Competing interests

The authors declare no competing interests.
