## [Peer Review File · Nature Communications]

Reviewers' Comments:

Reviewer #1:

Remarks to the Author:

This is a review of the manuscript entitled "A shared neural code for the physics of actions and object events" by Seda Akbiyik, Alfonso Caramazza and Moritz Wurm.

The Authors report a functional MRI study in which they sought to identify the neural representations of actions performed by animate and non-animate entities. The study extends previous work on the neural mechanisms associated with recognition of different kinds of actions by directly comparing, in a within-subject design, the neural representations of structurally similar kinds of actions (or events, to use the Authors' preferred wording) performed by human actors or by objects interacting with each other. A battery of video stimuli depicting three kinds of actions/events (walk-jump-kick for human actions; roll-bounce-hit for object events) was used to evoke BOLD responses that were assessed using state-of-the-art multivariate activation pattern analysis techniques. The search for animacy-independent and stimulus modality-independent representations were adequately implemented using classifier cross-decoding (including pair-wise cross-decoding). The impact of action/event-irrelevant stimulus features was minimised by using stimuli varying in viewpoints, subjects, animate/inanimate passive patients, and moving directions. The results revealed that brain regions classically associated with action observation (lateral occipitotemporal cortex - LOC, ventral premotor cortex, and anterior inferior parietal lobe) contain a shared neural code for actions and object events; that LOC carries representations invariant to stimulus modality, and that right temporal and parietal regions are more sensitive to actions performed by animate agents.

This is an excellent manuscript. The aim of the study is clearly stated, the methods (experimental design, number of participants, participant task, analysis of behavioural data, fMRI data recording and analysis techniques, thresholding used) seem well suited to address the study aims, the results are clearly reported, and the findings of the study contribute to advancing knowledge in the research domain.

The main point I would like the Authors to develop is their discussion of what really is coded in the neural representations that their cross-animacy-decoding analyses uncovered. The structural similarity between the animate actions and object events allowed the identification of "event representations that are commonly defined by both actions and object events." What do these representations really contain, i.e. what do they code, and in what way do they support human mental activities? The Authors attempted to explore these contents using an RSA analysis (SuppFig 4), which revealed that "large parts of the action observation network, particularly left anterior IPL, are sensitive to inter-object relations such as making contact, whereas motion trajectory was captured in dorsal occipital and medial premotor cortices (see Supplementary Figure 4). [...] these initial findings suggest that parts of the action observation network may encode events at a level specifying inter-object relations." In the Discussion, the Authors state that "these regions [IPL and premotor cortex] might be more properly construed as representing the physics and kinematics of events even if they lack any motor-relevant properties", and later, "Left LOTC seems to encode events at a sufficiently abstract level that can capture such semantic relations [e.g. make contact with, cause] between agents, objects, and their environment." These conclusions seem reasonable given the results, but I still feel somewhat unsatisfied as to what this really means. Such cross-animacy representations may also be implicated in findings from social cognition, i.e. the recruitment of neural representations of mental states by both interacting humans and abstract interacting objects (e.g. animations of the type classically studied by Heider and Simmel). More speculatively, such representations may explain the human propensity to anthropomorphise interpretations of all kinds of non-human behaviour. While these are of course speculations, I would like to encourage the Authors to venture a little further on how the representations they uncovered contribute to the way humans interpret events around them.

Minor comments

To increase results parsing, I would recommend structuring Figures 3 and 4 in a more similar way

(i.e. separation between whole-brain results in A and ROI findings in B used in Figure 3 could be also used in Figure 4).

Figure 4: Figure structure: legend refers to "bottom row", but there's no "top row".

Page 5, line 119: "accuracies from six independently defined ROIs in each hemisphere". That sentence reads to me as if 12 ROIs were investigated in total, while there were 6 in total - 3 per hemisphere. I suggest revising that wording.

Page 6, line 157: "To sum, ..." I would recommend replacing with "In sum".

There are a few typos that would warrant a systematic search for similar ones in the manuscript, e.g. "...Wurm et al., 2017). the actions..." (page 4, line 113).

Reviewer #2:

Remarks to the Author:

This study asks the interesting question of whether the brain regions that respond during viewing of actions of animate entities (i.e., the fronto-parietal and posterior temporal action observation network) also carry information about object events. Further, the study also asks if the representation of event information is invariant to the stimulus modality (visual vs linguistic). The study reports that i) regions of the "action observation network" encode the actions of agents and objects in a common fashion, but ii) only the left lateral occipitotemporal cortex encodes information about events that generalized across both animacy and modality, and iii) right STS and TPJ and bilateral parietal lobes respond more to actions of agents than to object events.

The study asks an interesting question and uses appropriate analysis methods to answer it. However, there are a few significant concerns regarding the stimulus design and theoretical background that dampen enthusiasm.

Main comments:

1. The motivation of the study seems to lack a clear theoretical focus, and the three main findings listed above feel like a bit of a grab-bag. Is the "action observation network" really a Thing, or is it just a phrase that has been used in a small number of prior papers to refer to an ill-defined and widespread set of brain regions that respond when viewing actions but may do lots of other things and probably differ from each other? If the "action observation network" is not an empirically or theoretically coherent clear Thing, then the question of the nature of the representations it holds is less theoretically motivated. To me, this paper read as a kind of exploratory study which is interesting, reasonable and publishable, but perhaps does not quite reach the level expected of Nature Communications.

2. Although the authors have tried to control for some low-level visual features by using different motion directions and perspectives, it looks like actions and object events can be decoded from the occipital regions (including primary visual cortex; Figures 2A & 2B) which would suggest that there might be low-level visual features that are confounded in the stimulus design. This is also a concern for the generalization results showing decoding of events invariant to animacy (Figure 3A) – posterior occipital areas seem to show greater than chance cross-decoding accuracies. It will be useful to have the ROI analysis for Early Visual areas (V1/V2) to rule out potential (low-level) visual confounds.

Even the weaker but significant cross-decoding (animate to inanimate) of verbally described events in the posterior occipital regions also points towards a potential visual feature confound (Figure 4A). Authors should report the visual extent of the words on the display, and average word-length of each sentence. It seems unlikely that there would be systematic differences in visual statistics of sentences across animate and inanimate that is preserved across animacy, but it will be good to rule out the possibility.

3. The cross-animacy + cross-modality decoding analysis uses a classifier trained on videos of one condition (action events) and tested on sentences of another condition (object events). This analysis jumps a step ahead by not showing the cross-modal within animacy decoding (i.e., train on videos of action events and test on sentences of object events, and vice-versa). This should be a stronger effect than the cross-animacy + cross-modality decoding, and if not, it would point to some possibly spurious correlations in the stimuli that is leading to the observed effect only in LOTC (Figure 4B). It would be difficult to interpret the result in such a case, but cross-animacy + cross-modality decoding (or lack thereof) in V1/V2 should at least rule out any (unlikely) cross-modal visual feature confounds.

Other comments:

- One possible explanation for cross-animacy + cross-modal decoding could be mental imagery. Although not crucial for the interpretation of the results, it would be illuminating to check for imagery by testing for order effects in cross-modality generalization. Presumably, watching sentences after videos would lead to more vivid imagery and yield stronger effect sizes than reading the sentences before watching the videos.
- The comparison in Figure 2C (Action > Object event decoding) seems conceptually flawed. Action event and object event decoding can be easy or difficult for many reasons, and one can make them arbitrarily different by systematically modifying some stimulus properties. For example, one can make it trivially easier to decode action events by making the agents bigger compared to the corresponding objects in the object event conditions. This would lead to bigger pixel-level changes in successive frames of the videos for the action events, leading to distinct and easy to decipher motion trajectories, and therefore much easier understanding of the event unfolding in the video. One way to address this concern is by collecting human behavioral ratings on event recognition and showing that both action and object event recognition are equally difficult for humans on the stimuli used in the study.

Reviewer #3:

Remarks to the Author:

This manuscript presents a timely investigation of the neural substrates of action events that encompass both animate and inanimate agents, regardless of the modality of presentation of events (visual or verbal). This is an important issue following a long tradition of research on the role of the Action Observation Network (AON) in action understanding. As a consequence of motor mirroring, the involvement of the AON in the identification of others' actions has been assumed to be specific to human actions. The present study demonstrates that some regions of the AON, in particular the lateral occipito-temporal cortex, code for the perception of action events that generalize across animate (human) and inanimate (object) agents.

Multivariate pattern analysis techniques are applied on fMRI data from 25 participants. After observing the brain regions activated during the perception of each type of event, different levels of cross-decoding procedures are conducted to identify the brain regions that generalize over animacy and stimulus modality. The experiment design is elegant and the study rigorously conducted.

My comments and suggestions mostly aim at stressing the novelty of the findings and delineate their scope. I also question a few methodological choices. I detail my comments below.

1) The study should be better articulated with the theory developed by Jeff Zacks on Event Perception and Representation, which also aims at generalizing across different types or formats of events. A recent review paper is briefly cited on page 1 of the Introduction, but I think that the theoretical issues raised by Zacks et al. and the empirical evidence provided by their work are highly relevant for the current study and deserve more consideration. The similarities and divergences with this line of research should be clearly highlighted throughout the manuscript.

2) The events and the actions investigated may be mostly differentiated based on motion trajectories. I understand that this is related to the need to have similar structures in the design of animate/inanimate events, but I believe this is an important specificity to consider and discuss. Should we expect the same degree/locus of common neural substrate for other types of events

that do not involve that much motion (e.g. multistep functional events, human using objects versus automatic machine)? For example, one may expect predictive processes to be involved in the understanding of motion events. Variations in predictability of the different motion trajectories might explain the high degree of overlap in the neural substrate of event perception in the animate and inanimate conditions (especially when the task, via the catch-trials, orients towards prediction).

3) The critical role of the common node identified should be further discussed. Is it possible that the areas identified do not have the same critical status for the different conditions, despite involving the same neural network? To illustrate with a putative scenario, verbal stimuli might lead to visual simulation of the verbal content, but this simulation might not be essential for the understanding of the sentence. Left LOTC might be therefore critical for visual processing of events but simply a by-product of the processing of event sentences.

4) The analyses logically focus on the "core regions of the action observation network". Yet ROI selection is not sufficiently justified. On page 16 section ROI Analysis, the authors explain that selection was based on the very nice meta-analysis by Casper et al. (2010) but the regions identified as part of the AON in Casper et al. have not all been shortlisted. In particular, I am very surprised that the ROI analyses did not consider SPL and pMTG. This is even more puzzling that SPL and STS are initially highlighted in the distinct neural representations of actions and object events on page 6. In parallel, I encourage the authors to use a very precise and consistent definition of the AON throughout the paper, as the term does not mean much otherwise.

5) The discussion of the main findings, in particular the left/right asymmetry, may also include the role of the methodology used, which is never addressed. For example, I wonder if there is a relation between the overall amount of activity in a given region/hemisphere and the chance of successful cross-decoding in this region/hemisphere. On a related note, would the result about action/event generalization (p. 7 and Fig 3) be stronger with the addition of a control ROI that does not show generalization across animacy?

6) Finally, I list below a few more minor suggestions:

- Presentation of ROI results (p. 6): the presentation of the different contrasts is difficult to follow. I recommend to decompose the interaction in the same manner for left and right hemispheres.
- Sample size (p. 12): I know that there is no actual real standard for apriori power analyses in this type of neuroimaging study, but I think it is still good practice to justify the sample size considered with the best that can be found.
- Stimuli (p.13): I encourage the authors to provide an illustration of the different possible exemplars for one event as supplementary materials so that the reader can appreciate their variability.
- Figure captions: I do not see what is "outlined in black" on the maps.
- Catch-trials (p. 14): the choice of catch-trials is very important because it defines the task participants actually perform. Were there neural differences between the conceptual vs. perceptual catch-trials? I could not figure out how the different catch trials were organized within blocks/sessions.
- Event classification (p. 15): I had a hard time understanding to which conditions the two predictors "based on 4 trials selected from the odd and 4 selected from the even trials of a run" refer to. Is it the verbal/ visual modality?
- ROI analysis (p.17): Is it possible to add random slopes for items in the mixed-models used?

References

- Zacks J. M. (2020). Event Perception and Memory. *Annual review of psychology*, 71, 165–191. <https://doi.org/10.1146/annurev-psych-010419-051101>
- Radvansky, G. A., & Zacks, J. M. (2011). Event perception. *Wiley interdisciplinary reviews. Cognitive science*, 2(6), 608–620. <https://doi.org/10.1002/wcs.133>
- Franklin, N. T., Norman, K. A., Ranganath, C., Zacks, J. M., & Gershman, S. J. (2020). Structured

Event Memory: A neuro-symbolic model of event cognition. *Psychological review*, 127(3), 327–361. <https://doi.org/10.1037/rev0000177>

Richmond, L. L., & Zacks, J. M. (2017). Constructing Experience: Event Models from Perception to Action. *Trends in cognitive sciences*, 21(12), 962–980. <https://doi.org/10.1016/j.tics.2017.08.005>

Zacks, J. M., Braver, T. S., Sheridan, M. A., Donaldson, D. I., Snyder, A. Z., Ollinger, J. M., Buckner, R. L., & Raichle, M. E. (2001). Human brain activity time-locked to perceptual event boundaries. *Nature neuroscience*, 4(6), 651–655. <https://doi.org/10.1038/88486>

A note to Reviewers

We thank the reviewers for their thoughtful comments on the original submission. We have considered each of the comments raised and have made substantial revisions to the manuscript. In the process of revising the manuscript, we discovered that the order of functional runs was corrupted in seven subjects resulting in a wrong assignment of logfiles for the sentence session and nonoptimal co-registration. We fixed this issue, carefully checked our complete analysis pipeline for correctness, and re-ran all our analyses. We would like to note that this fix reduced the noise in our results, made the within-sentence and cross-modal effects stronger, and did not affect our overall conclusions. We apologize for any inconvenience this update might cause and for not having identified this issue earlier. We detail the associated changes below:

- Same results for the video session except for subtle changes in the group maps that do not change the implicated regions or our conclusions (see Figures 2 & 3).
- Cross-animacy (verbal): the ROI analysis yielded the same results as our prior findings (see Figure 4c). In the whole-brain, the effects get stronger, and regions that only showed up in the ROI analysis now survive correction for multiple comparisons in the whole-brain (e.g., left ventral premotor cortex, inferior parietal lobe, see Figure 4a). Notably, these updates do not contribute to the interpretation of our findings.
- Cross-animacy + modality: In the whole-brain, the cluster in left LOTC is now stronger and more well-defined, and the effects extend to ventral temporal cortex, and some right posterior temporal clusters (see Figure 4b). Reviewer 2 requested that we elaborate on cross-animacy + modality generalization in posterior occipital regions. We address these effects via the associated revisions in p.11. In the ROI analysis, not only left LOTC but also left IPL showed above-chance cross-animacy + modality generalization (see Figure 4d). We address this new effect in p.12.

Below are our point-by-point responses to each of the Reviewers' comments, and the associated revisions we made to the manuscript.

Responses to Reviewers' comments

Reviewer #1 (Remarks to the Author):

This is a review of the manuscript entitled "A shared neural code for the physics of actions and object events by Seda Akbiyik, Alfonso Caramazza and Moritz Wurm.

The Authors report a functional MRI study in which they sought to identify the neural representations of actions performed by animate and non-animate entities. The study extends previous work on the neural mechanisms associated with recognition of different kinds of

actions by directly comparing, in a within-subject design, the neural representations of structurally similar kinds of actions (or events, to use the Authors' preferred wording) performed by human actors or by objects interacting with each other. A battery of video stimuli depicting three kinds of actions/events (walk-jump-kick for human actions; roll-bounce-hit for object events) was used to evoke BOLD responses that were assessed using state-of-the-art multivariate activation pattern analysis techniques. The search for animacy-independent and stimulus modality-independent representations were adequately implemented using classifier cross-decoding (including pairwise cross-decoding). The impact of action/event-irrelevant stimulus features was minimized by using stimuli varying in viewpoints, subjects, animate/inanimate passive patients, and moving directions. The results revealed that brain regions classically associated with action observation (lateral occipitotemporal cortex - LOC, ventral premotor cortex, and anterior inferior parietal lobe) contain a shared neural code for actions and object events; that LOC carries representations invariant to stimulus modality, and that right temporal and parietal regions are more sensitive to actions performed by animate agents.

This is an excellent manuscript. The aim of the study is clearly stated, the methods (experimental design, number of participants, participant task, analysis of behavioural data, fMRI data recording and analysis techniques, thresholding used) seem well suited to address the study aims, the results are clearly reported, and the findings of the study contribute to advancing knowledge in the research domain.

Comment: The main point I would like the Authors to develop is their discussion of what really is coded in the neural representations that their cross-animacy-decoding analyses uncovered. The structural similarity between the animate actions and object events allowed the identification of "event representations that are commonly defined by both actions and object events." What do these representations really contain, i.e., what do they code, and in what way do they support human mental activities? The Authors attempted to explore these contents using an RSA analysis (SuppFig 4), which revealed that "large parts of the action observation network, particularly left anterior IPL, are sensitive to inter-object relations such as making contact, whereas motion trajectory was captured in dorsal occipital and medial premotor cortices (see Supplementary Figure 4). [...] these initial findings suggest that parts of the action observation network may encode events at a level specifying inter-object relations." In the Discussion, the Authors state that "these regions [IPL and premotor cortex] might be more properly construed as representing the physics and kinematics of events even if they lack any motor-relevant properties", and later, "Left LOTC seems to encode events at a sufficiently abstract level that can capture such semantic relations [e.g., make contact with, cause] between agents, objects, and their environment." These conclusions seem reasonable given

the results, but I still feel somewhat unsatisfied as to what this really means. Such cross-animacy representations may also be implicated in findings from social cognition, i.e., the recruitment of neural representations of mental states by both interacting humans and abstract interacting objects (e.g., animations of the type classically studied by Heider and Simmel). More speculatively, such representations may explain the human propensity to anthropomorphise interpretations of all kinds of non-human behaviour. While these are of course speculations, I would like to encourage the Authors to venture a little further on how the representations they uncovered contribute to the way humans interpret events around them.

Response:

We thank the Reviewer for their kind comments, and for encouraging us to elaborate further on the nature of cross-animacy representations that we have uncovered.

In the manuscript, we entertained the proposal that shared visual and kinematic information – the “physics” of visual scenes – and high-level semantic information are the features that are being captured by these cross-animacy representations. More broadly, we argued that these cross-animacy representations encode event properties that go beyond the animacy of the entities that are involved. Even as we tried to go beyond speculation and identify what is being encoded by these cross-animacy representations with our RSA analyses, these analyses were necessarily constrained by the nature of our stimuli.

In our discussion of cross-animacy generalization, we put emphasis on shared representations of events that can apply to both movements of humans and inanimate entities. However, as the Reviewer points out, another plausible interpretation of the results is that perceivers ascribed human-like attributes to inanimate objects to make sense of their movement (Heider & Simmel, 1944; Michotte, 1946; Scholl & Tremoulet, 2000). Based on this argument, cross-animacy generalization would happen not due to encoding of a general description of events regardless of animacy, but rather, because human-like properties are attributed to object events.

We agree with the Reviewer that the cross-animacy representations we identified might have partly been driven by the attribution of human-like properties to the movements of objects. However, we would like to emphasize that we used filmed real-world movements of objects instead of abstract animations. Thus, there was no question as to the inanimate nature of the objects in the object event condition, nor did the objects move in ways that might give the impression of animacy or intentionality akin to Heider & Simmel paradigms (e.g., autonomous motion that appears to be goal-directed). Still, perceivers could attribute human-like characteristics to non-human entities even when those entities do not move in ways that signal animacy. Identifying the extent of such effects would be an interesting challenge for future

studies but would require assuming that participants can process inanimate entities as animate against evidence about their inanimate nature.

We thank the Reviewer for encouraging us to elaborate on these issues, and in response to their comments, we now changed some parts of the discussion and present the following paragraph at p. 14.

“Our findings point to new directions that will shed light on how the brain represents dynamic information about agents and objects. An avenue for future work is identifying which shared aspects of actions and object events are encoded by their common neural representation in frontoparietal and posterior temporal cortices. Candidate features include, but are not limited to, shared visual and kinematic information, physics of visual scenes, and high-level semantic information. Another alternative is that the propensity to attribute human-like characteristics to inanimate entities, *anthropomorphizing*, underlies the common neural representation of actions and object events that we have uncovered (Heider & Simmel, 1944; Michotte, 1946; Scholl & Tremoulet, 2000). Spontaneous recruitment of regions involved in social cognition while reasoning about movements of simple visual shapes supports this possibility (e.g., Castelli et al., 2002; Gobbin et al., 2007, Wheatley et al., 2007). According to this alternative, the cross-animacy representations we uncovered would not necessarily reflect encoding of event components that are shared across actions and object events, but rather, encoding of human-like properties that are attributed to objects’ movement. Although possible, we think that this is highly unlikely in the context of our experiment since we used filmed events. Thus, there was no question as to the inanimate nature of the objects in the object event condition, nor did they move in ways that might signal animacy or intentionality (e.g., autonomous motion that appears to be goal-directed). Still, addressing the contributions of anthropomorphic reasoning to the shared neural representations of actions and object events will be an important challenge for future work.”

Minor comments

Comment: To increase results parsing, I would recommend structuring Figures 3 and 4 in a more similar way (i.e., separation between whole-brain results in A and ROI findings in B used in Figure 3 could be also used in Figure 4).

Response: We implemented this change.

Comment: Figure 4: Figure structure: legend refers to “bottom row”, but there’s no “top row”.

Response: Corrected.

Comment: Page 5, line 119: "accuracies from six independently defined ROIs in each hemisphere". That sentence reads to me as if 12 ROIs were investigated in total, while there were 6 in total - 3 per hemisphere. I suggest revising that wording.

Response: Corrected. Note that we now added two new ROIs (superior temporal sulcus and superior parietal lobe) as per Reviewer 3's request. The paragraph on ROI selection has been changed as shown below.

“To obtain a better understanding of action and object event decoding across specific regions, we extracted classification accuracies from independently defined regions of interest (ROIs) in each hemisphere. We primarily focused on regions of the action observation network that are most strongly and consistently recruited during action observation tasks: lateral occipitotemporal cortex (LOTc), ventral premotor cortex (PMv), and inferior parietal lobe (IPL) (for reviews, see Caspers et al., 2010; Molenberghs et al., 2012). To provide a more fine-grained picture of how actions and object events are represented in other areas that are also linked to action observation, we also report ROI results from superior parietal lobe (SPL) and superior temporal sulcus (STS) (see Methods for more details on ROI selection).” (p. 6).

Comment: Page 6, line 157: "To sum, ..." I would recommend replacing with "In sum".

Response: Replaced.

Comment: There are a few typos that would warrant a systematic search for similar ones in the manuscript, e.g., "...Wurm et al., 2017). the actions..." (page 4, line 113).

Response: Thank you for bringing these to our attention. We have made the appropriate correction (the dot should have been a comma) and made sure that there are no similar typos throughout the manuscript

Reviewer #2 (Remarks to the Author):

This study asks the interesting question of whether the brain regions that respond during viewing of actions of animate entities (i.e., the fronto-parietal and posterior temporal action observation network) also carry information about object events. Further, the study also asks if the representation of event information is invariant to the stimulus modality (visual vs linguistic). The study reports that i) regions of the "action observation network" encode the actions of agents and objects in a common fashion, but ii) only the left lateral occipitotemporal cortex encodes information about events that generalized across both animacy and modality,

and iii) right STS and TPJ and bilateral parietal lobes respond more to actions of agents than to object events.

The study asks an interesting question and uses appropriate analysis methods to answer it. However, there are a few significant concerns regarding the stimulus design and theoretical background that dampen enthusiasm.

Comment: The motivation of the study seems to lack a clear theoretical focus, and the three main findings listed above feel like a bit of a grab-bag. Is the "action observation network" really a Thing, or is it just a phrase that has been used in a small number of prior papers to refer to an ill-defined and widespread set of brain regions that respond when viewing actions but may do lots of other things and probably differ from each other? If the "action observation network" is not an empirically or theoretically coherent clear Thing, then the question of the nature of the representations it holds is less theoretically motivated. To me, this paper read as a kind of exploratory study which is interesting, reasonable and publishable, but perhaps does not quite reach the level expected of Nature Communications.

Response:

We agree with the Reviewer that the 'action observation network' encompasses a wide range of regions that potentially have distinct roles in serving action recognition and understanding. We also agree with the Reviewer that there is some variation across researchers in terms of what specific regions this network entails. However, we would like to highlight that the referent of the phrase 'action observation network' is not as ill-defined or vague as the Reviewer suggests. A quick google scholar search shows that the phrases 'action observation network' or 'action observation system' alone appear in around 2500 papers (see links below). Although action observation recruits a wide range of areas throughout frontoparietal and occipitotemporal cortices, these phrases are commonly used to refer to three main regions: inferior parietal lobe, ventral premotor cortex/inferior frontal gyrus, and posterior temporal cortex.

https://scholar.google.com/scholar?hl=en&as_sdt=0%2C22&q=%22action+observation+network%22&btnG=

https://scholar.google.com/scholar?hl=en&as_sdt=0%2C31&q=%22action+observation+system%22&btnG=

Furthermore, the frontoparietal component of the action observation network is also commonly referred to as the “mirror neuron system”. A quick google scholar search shows that the phrase ‘mirror neuron system’ alone appears in around 30,200 papers. Although not all researchers consider the posterior temporal cortex as a part of the mirror neuron system, there is consensus that the mirror neuron system receives inputs from it.

https://scholar.google.com/scholar?hl=en&as_sdt=0%2C31&q=%22mirror+neuron+system%22&btnG=

Putting these notes aside, we agree with the Reviewer that it is important for us to clarify what we mean by the Action Observation Network (AON) throughout the manuscript, as also pointed out by Reviewer 3. We implemented various changes throughout the manuscript to clarify what we mean by AON. To iterate them here, we had focused on three main regions that are most strongly and consistently activated in action perception studies: lateral occipitotemporal cortex, ventral premotor cortex, and inferior parietal lobe (Caspers et al., 2010; Molenberghs et al., 2012). Lesions to these three main regions are also associated with deficits in action understanding (Urgesi et al., 2014). As per Reviewer 3’s request, we now added superior temporal sulcus and superior parietal lobe to our ROI analyses as well.

We referred to the AON as a placeholder for frontoparietal and posterior temporal brain regions that are implicated in action understanding, and that were specifically addressed in previous work in relation to their role in understanding human actions. However, putting the exact referent of the AON aside, we do not think the implications of our results are necessarily tied to the action observation network being a “*Thing*”. The aim of our study was to reveal the shared and distinct neural representations of actions that involve animate agents and events that involve inanimate objects, which raises interesting theoretical questions on its own right. In the original manuscript, we had started the Introduction by briefly addressing these theoretical issues. We connected our motivations and findings to the literature on the AON to highlight that it is typically assumed that these regions support recognizing actions of animate entities.

In response to the Reviewer’s comments, we now made various changes in the Introduction and throughout the manuscript to make sure that what we mean by the ‘Action Observation Network’ is now clearer and that the readers can appreciate the theoretical motivations behind our study over and above AON being a “*Thing*”. See the changes in the opening paragraphs of the Introduction for a clearer description of the AON and the theoretical motivation behind the study (p. 2). The paragraph we added for clarifying ROI selection in response to Reviewer 3’s request is also relevant for this purpose (see p. 6).

Comment: Although the authors have tried to control for some low-level visual features by using different motion directions and perspectives, it looks like actions and object events can

be decoded from the occipital regions (including primary visual cortex; Figures 2A & 2B) which would suggest that there might be low-level visual features that are confounded in the stimulus design. This is also a concern for the generalization results showing decoding of events invariant to animacy (Figure 3A) – posterior occipital areas seem to show greater than chance cross-decoding accuracies. It will be useful to have the ROI analysis for Early Visual areas (V1/V2) to rule out potential (low-level) visual confounds.

Response: In the video condition, we find above-chance action decoding, object event decoding, and cross-animacy generalization in a wide range of areas, including, as the Reviewer points out, in occipital brain regions. Since the original figures did not show the full occipital lobes, below are whole-brain maps showing occipital pole as well as medial occipital regions for action decoding, object event decoding, and cross-animacy decoding in the video session.

Figure 1. Action decoding, object-event decoding, and cross-animacy decoding of events in the video condition. The maps are thresholded by areas corrected for multiple comparisons using Monte Carlo Cluster based correction ($p_{\text{initial}} = .001$).

The maps above (Figure 1) are thresholded by areas corrected for multiple comparisons, thus, all the colored portions show above-chance decoding. From this display, it can be seen that there is above chance decoding in early visual areas and throughout the occipital lobe for all three decoding schemes. To address the Reviewer's point more directly, we also ran an additional ROI analysis on the cross-animacy video session (Figure 2). We used the centroid Talairach coordinates of Brodmann area 17 for left and right primary visual cortices following Brodmann definitions provided by Lacadie et al., (2008) - TAL coordinates: right primary visual [11 -75 11], left primary visual [-11 -78 10]. We created spheres with a 12 mm radius around these coordinates as we did for all our ROI definitions. Note that we are now showing the results for bilateral superior parietal lobes (SPL) and superior temporal sulcus (STS) as per Reviewer 3's

request. Consistent with the whole-brain results, this analysis revealed above-chance cross-animacy decoding in bilateral primary visual cortices in the video session.

Figure 2. Additional ROI analyses of cross-animacy decoding of events, video condition. Error bars indicate standard error of the mean (SEM), and asterisks indicate FDR-corrected effects: * $p < .05$, ** $p < 0.01$, *** $p < 0.001$, **** $p < 0.0001$.

Throughout the paper, we do not disregard the possibility that the low-level visual properties of events might be part of what is being reflected in the action or object event decoding or decoding of events across animacy. Our aim in introducing perceptual variance was to make sure that we do not train the classifiers purely on low level visual properties, and that they generalize at least over certain low-level properties such as moving direction, viewpoint, and specific actors or objects. However, we do not think, or have claimed that our event categories are immune to variation in low-level visual features.

In addition, we would like to point out that it is not clear how this raises a theoretical concern for our interpretations. First, let us assume that cross-animacy decoding in occipital lobes is driven solely by variation in low-level visual properties. This does not necessarily mean that decoding in frontoparietal regions and temporal cortex more anteriorly also reflect encoding of such properties. The possibility that decoding in early visual regions might capture variation in low-level visual features does not explain why we observe decoding in higher-level brain regions or why decoding accuracies in most of these higher-level regions (e.g., LOTC, STS, SPL, IPL) are higher than in primary visual cortex.

Second, and perhaps more importantly, differences in visual properties across event categories are part of what defines their meaning. Regardless of how much perceptual variance one might induce, a *jump-bounce* event will always involve vertical movement while a *walk-roll*

event does not have to. Vertical motion is part of what defines 'jump' or 'bounce'. Similarly, even if we were to introduce more perceptual variance, the 'kick' and 'hit' events would elicit tracking of two interacting objects while the jump-bounce and walk-roll events could be processed by tracking movements of one object alone. It is practically impossible to rule out variation in all visual properties when comparing different motion events that are coherent and meaningful.

We thank the reviewer for encouraging us to address the effects we observe throughout the occipital lobe and address the contribution of low-level visual properties to decoding of events. In response to the Reviewer's comments, we now emphasize more strongly the potential role of low-level visual properties of events in driving our decoding analyses and point out the successful decoding throughout early occipital regions (see the changes in pp. 7, 9). The paragraph we added in p. 14 in response to Reviewer 3's comment on generalization of these findings to other event types that are not primarily defined by motion path may also speak to this issue.

Comment: Even the weaker but significant cross-decoding (animate to inanimate) of verbally described events in the posterior occipital regions also points towards a potential visual feature confound (Figure 4A). Authors should report the visual extent of the words on the display, and average word-length of each sentence. It seems unlikely that there would be systematic differences in visual statistics of sentences across animate and inanimate that is preserved across animacy, but it will be good to rule out the possibility.

Response: We thank the Reviewer for this comment and will unpack it below.

We agree that the cross-animacy decoding in the sentence session (Figures 4a,c) could have relied on visual or word-length differences across sentences, which would be orthogonal to their meaning. This is one of the reasons why we do not base any of our overall conclusions on this analysis. Also, as it can be seen from the following phrase in the original manuscript, we mention sentence length or syntactic differences across events as contributing factors to cross-animacy decoding in the sentence session as soon as we reported these results.

“Successful cross-decoding in the sentence session might have been driven by certain stimulus characteristics such as sentence length or presence/absence of prepositions that do not pertain to the meaning of the event per se.”

We had already reported the different words we used for sentence stimuli in the stimulus section of our manuscript. Based on the Reviewer's request, and for ease of comparison across categories, we now also added a table with sentences and their respective word length in our supplementary files (<https://osf.io/3wrv5>). A quick comparison of different sentences shows that the verb phrases across the three events were not equal in terms of the number of words, or

word length. As it can be seen from the manuscript as well as this table, while the ‘hit-kick’ events did not contain a preposition and were shorter for both actions and object events, the ‘walk/roll in front’ and ‘jump/bounce over’ events contained a preposition and were longer.

We would like to emphasize that the evidence we provide with the sentence data was complementary to the video evidence, and due to the limited nature of events that are structurally similar across actions and object events, we ignored such syntactic and sentence-level differences while picking our stimuli. Again, we had addressed this issue as it can be seen from the following quote in the original manuscript:

“Since our analyses on the shared aspects of actions and object events relied on generalization across stimulus types, perceptual and syntactic differences between sentences, such as sentence length and presence/absence of prepositions, were ignored.”

To make sure that this message is delivered in a clearer fashion, we now made several changes in the manuscript emphasizing successful cross-animacy generalization of sentence stimuli throughout the occipital lobe, and the possibility that certain perceptual and syntactic differences might have contributed to these findings (see pp. 10-11).

In the rest of this comment, the Reviewer made the following statement: “It seems unlikely that there would be systematic differences in visual statistics of sentences across animate and inanimate that is preserved across animacy, but it will be good to rule out the possibility.”

The underlined parts of the sentence shown above mean the same thing: ‘systematic differences across animate and inanimate, preserved across animacy’. We are not sure if the second part of the sentence was an error in writing and the Reviewer just meant to say, ‘systematic differences across animate and inanimate’ or their aim was to say, ‘It seems unlikely that there would be systematic differences in visual statistics of sentences **across animate and inanimate** that is preserved across **modality**’.

To make sure that we capture both options, below are the requested ROI analyses for the whole-brain maps reported in Figures 4a-b of the manuscript: cross-animacy verbal and cross-animacy + modality decoding. Again, we also report STS and SPL as per Reviewer 3’s request.

Figure 3. Additional ROI analyses for cross-animacy decoding of events in the sentence session and cross-animacy + modality decoding. Error bars indicate standard error of the mean (SEM), and asterisks indicate FDR-corrected effects: * $p < .05$, ** $p < 0.01$, *** $p < 0.001$, **** $p < 0.0001$.

Primary visual cortex shows above chance decoding across animacy in the sentence session, but it does not show generalization across both animacy and modality. Successful cross-decoding of events in the sentence session in the primary visual cortex is not surprising given the differences we addressed above. And we hope that the lack of significant decoding in the primary visual cortex for cross-animacy + modality decoding alleviates some of the Reviewer’s concerns.

We would like to note that we completed these analyses to address the Reviewer’s comments, but we have some concerns about the logic of this ROI analysis. Following the Reviewer’s reasoning, the absence of decoding in primary visual cortex for cross-animacy decoding in sentences, or cross-animacy + modality decoding could point to lack of a visual feature confound. However, a region might fail to show decoding for a variety of reasons. That we do not show cross-animacy + modality decoding in the primary visual cortex does not eliminate the possibility of low-level visual confounds in the stimuli. Our analysis method might not be sensitive enough to capture very subtle effects, which might lead to lack of decoding. On the other hand, even if we showed cross-animacy + modality decoding in these regions, this would not guarantee that such effects were due to encoding of low-level visual properties. To push this logic even further, even if we had shown cross-animacy + modality generalization in primary visual cortex and knew for sure that generalization there happened due to visual confounds, this still wouldn’t explain why we observe generalization in regions that are not tied to encoding of low-level visual features (e.g., superior temporal sulcus, lateral occipitotemporal cortex).

Altogether, even though we are happy to provide the Reviewer with the requested analyses, given the concerns raised above, we decided not to include these analyses in the main

manuscript. However, we agree that it is vital for us to point out the effects we observe in posterior occipital regions and raise the possibility that there might be some perceptual factors that generalize across animacy, or modality. We made several changes throughout the Results and Discussion sections of the manuscript to address these issues (see changes in pp. 11-12).

Comment: The cross-animacy + cross-modality decoding analysis uses a classifier trained on videos of one condition (action events) and tested on sentences of another condition (object events). This analysis jumps a step ahead by not showing the cross-modal within animacy decoding (i.e., train on videos of action events and test on sentences of object events, and vice-versa). This should be a stronger effect than the cross-animacy + cross-modality decoding, and if not, it would point to some possibly spurious correlations in the stimuli that is leading to the observed effect only in LOTC (Figure 4B). It would be difficult to interpret the result in such a case, but cross-animacy + cross-modality decoding (or lack thereof) in V1/V2 should at least rule out any (unlikely) cross-modal visual feature confounds.

Response: In this comment, the Reviewer uses the following statement as their suggested new analysis “*train on videos of action events and test on sentences of object events, and vice-versa*”, but this describes the cross-animacy + modality analysis we already completed. We think that this was an error in writing, and that the Reviewer refers to the condition where the classifier is trained on videos of one condition and tested on the sentences of the same condition (train on videos of actions, test on sentences of actions), that is, cross-modal within-animacy decoding.

Below are the outputs of cross-modal within-animacy decoding presented alongside cross-animacy + modality decoding for comparison. As it can be seen from the figure below (Figure 4), at least for agent actions, cross-modal within-animacy decoding yielded stronger results compared to cross-animacy + modality decoding. However, we would like to note that comparison of decoding strength between cross-modal within-animacy versus cross-modal cross-animacy schemes is confounded by the number of samples used for training and testing. We agree with the reviewer that assuming equal number of samples for training and testing, cross-animacy + modality decoding should be weaker than cross-modal within-animacy decoding. However, in the context of our dataset, cross-modal within-animacy decoding only used half of the dataset while cross-modal cross-animacy benefits from the entire dataset. Putting this data analytical complexity aside, it is not clear how differences in accuracy for these two decoding schemes would address presence of spurious correlations. As we addressed above, decoding in posterior occipital regions could have been driven by such correlations, but this would not explain why we observe decoding in higher-level brain regions.

Figure 4. Cross-modal within-animacy decoding outputs along with cross-animacy + modality decoding. Whole-brain maps are thresholded by areas corrected for multiple comparisons using Monte Carlo Cluster based correction ($p_{\text{initial}} = .005$), except for cross-modal decoding of object events, which did not reveal strong effects. Error bars indicate standard error of the mean (SEM), and asterisks indicate FDR-corrected effects: * $p < .05$, ** $p < 0.01$, *** $p < 0.001$, **** $p < 0.0001$.

As it can be seen in Figure 4 presented above, the cross-modal within-animacy decoding analysis yielded weaker results for the object-event condition compared to the action condition. The reason for this difference is open to speculation, but we think that this might have to do with weaker results for object events in the sentence session compared to actions. To provide the full picture, below is a comparison of action and object event decoding within sentences and cross-modality (Figure 5).

Figure 5. Whole-brain three-way decoding of events **(A)** action sentences, **(B)** object event sentences, **(C)** actions across modality, and **(D)** object events across modality. For action or object event decoding across modality (C-D), a classifier is trained on videos of one event type (e.g., action videos) and tested on sentences of the same event type (e.g., action sentences). Maps in A-C are thresholded for areas that survive correction for multiple comparisons using Monte Carlo Cluster based correction ($p_{\text{initial}} = .005$). The map in D (cross-modal decoding of object events) is thresholded at $p < .05$ to demonstrate trends that do not survive correction. **(E)** ROI decoding accuracies for action and object event sentences. **(F)** ROI decoding accuracies for actions and object events across modality. Error bars indicate SEM and asterisks indicate FDR-corrected effects: * $p < .05$, ** $p < 0.01$, *** $p < 0.001$, **** $p < 0.0001$.

As it can be seen in this figure, both decoding of object events in the sentence session, as well as cross-modal decoding of object events were weaker compared to actions. There could be various reasons for this difference. First, the verbs or subjects we used for object event sentences might have resulted in more variable neural activity patterns compared to actions. Furthermore, weaker cross-modal decoding in the object event compared to action stimuli might imply that the visual-to-verbal reference is clearer and more specific in the context of actions, and more variable in the context of object events. Future studies with sentence stimuli that are more controlled in terms of their formal and visual properties could address these issues.

Overall, we thank the reviewer for encouraging us to report the cross-modal within animacy decoding outputs to give the readers a more complete understanding of various design properties and how they might impact decoding. We now added the cross-modal and within-sentence decoding of actions and object events to our supplementary information along with a discussion of these effects to give the readers the full picture (see Supplementary Figure 6).

Other comments

Comment: One possible explanation for cross-animacy + cross-modal decoding could be mental imagery. Although not crucial for the interpretation of the results, it would be illuminating to check for imagery by testing for order effects in cross-modality generalization. Presumably, watching sentences after videos would lead to more vivid imagery and yield stronger effect sizes than reading the sentences before watching the videos.

Response: We agree with the Reviewer that this definitely is a possibility. In fact, generalization across verbal and visual stimuli could be due to both visual imagery (in the sentence session) or verbalization (in the video session). As the Reviewer points out, we would expect stronger imagery in the participant group that started with the video session, and stronger verbalization in the participant group that started with the sentence session.

Below is the whole-brain contrast of the decoding maps of the two groups, which revealed no significant differences in areas that showed cross - animacy + modality decoding, which is demonstrated with the black outline (Figure 6). There was a trend for higher decoding in left STS for participants who received the sentence session first, which would be consistent with stronger verbalization in the video session, but this difference was not significant.

Figure 6. Decoding contrast of cross-animacy + modality based on session order. The black outline indicates areas that showed above-chance cross-animacy + modality decoding in the whole participant group.

Even though we did not find a clear order effect, it's still possible that cross-animacy + modality decoding was at least partly due to visual imagery. For instance, it is possible that the session order did not cause strong differences in the degree of imagery or verbalization, but that cross-animacy + modality generalization could still be underlain by it. To address concerns related to imagery or verbalization that were also partly raised by Reviewer 3, we now added the following paragraph on p. 11 and report the analysis of the order effect in our supplementary information (see Supplementary Figure 7).

“In the so-called action observation network, only left LOTC/STS showed significant cross-animacy + modality generalization that survived correction for multiple comparisons in the whole brain. Is this region causally involved in understanding events both through visual and verbal formats? If cross-animacy + modality generalization in LOTC is due to verbalization (in the video session) or visual imagery (in the sentence session), this might not be the case. For instance, if generalization across observed videos and sentences is due to imagery in the sentence session, this would not guarantee that neural activity during sentence processing is essential for understanding events in that modality. Since fMRI can only provide correlational evidence, evidence from individuals with brain damage or neuromodulation studies can be used to address the critical role of this region in understanding events through visual and verbal modalities. Still, our study allowed us to address the extent of imagery or verbalization effects as we balanced the order of video and sentence sessions across participants. If across modality generalization is due to imagery, we would expect stronger imagery for people who received the videos first and sentences second, and stronger verbalization for people who received sentences first and videos second. The contrast of the decoding maps of the two groups, however, revealed no significant differences indicative of an effect of imagery or verbalization (see Supplementary Figure 7). Thus, we found no support for the hypothesis that visual imagery or verbalization can fully account for the observed generalization across animacy and modality.” (p.11)

Comment: The comparison in Figure 2C (Action > Object event decoding) seems conceptually flawed. Action event and object event decoding can be easy or difficult for many reasons, and one can make them arbitrarily different by systematically modifying some stimulus properties. For example, one can make it trivially easier to decode action events by making the agents bigger compared to the corresponding objects in the object event conditions. This would lead to bigger pixel-level changes in successive frames of the videos for the action events, leading to distinct and easy to decipher motion trajectories, and therefore much easier understanding of the event unfolding in the video. One way to address this concern is by collecting human behavioral ratings on event recognition and showing that both action and object event recognition are equally difficult for humans on the stimuli used in the study.

Response: We agree with the Reviewer that the actions and object events we used in this study varied along many dimensions. Some of these dimensions were orthogonal to the animacy of the actor, for instance, as the Reviewer pointed out, the size of the moving entity. The main motivation for reporting differences in decoding strength across actions and object events was to show that despite these differences, we still show that actions and object events can be decoded in a remarkably similar way. Furthermore, with this analysis, we also hoped to reveal where in the brain information specific to actions or object events are encoded.

It is not clear to us what kind of a behavioral paradigm would help to address the issue the Reviewer is raising. Also, even if we came up with a clear behavioral paradigm to compare processing difficulty for actions and object events, and found out that they are not equally difficult, it is not clear how such a difference would affect our overall conclusions. Part of the difference could be due to differences in how easy it is to predict or recognize the unfolding of an event, and that might have to do with the ease with which we process actions of conspecifics compared to movements of objects. There could be a variety of factors related to attention or salience. Human movements might be inherently more interesting than moving objects, which could also explain differences in difficulty, if any. It is not clear to us what such differences would add to the overall story. Future research using animations to better control for low-level visual properties across actions and object events could speak to this issue.

One piece of evidence from our study that could speak to the point the Reviewer is raising is the performance in the catch trial task. As stated in the manuscript, in the video session, participants were equally likely to make a false alarm for actions and object events (Video: $t(24) = .86, p = .40, d = .18$). This, we think partly speaks to task demands for action and object event stimuli. Also, we now compared the catch trial detection accuracy for catch trials that had animate or inanimate actors, and found that participants were equally accurate in detecting a catch trial with a human or an object actor ($t(24) = .55, p = .59, d = .11$). We hope that this partially addresses any concerns about task difficulty giving rise to the effects we observed in the comparison of actions and object events.

To address the Reviewer's comments in the manuscript, we now updated the paragraph in our presentation of the comparison of action and object event decoding explaining the logic of this analysis in a clearer way and emphasize that the differences we observed could be partly due to certain differences between action and object stimuli that are orthogonal to the agency of the actor. See the updates in pp. 6-7.

Reviewer #3:

This manuscript presents a timely investigation of the neural substrates of action events that encompass both animate and inanimate agents, regardless of the modality of presentation of events (visual or verbal). This is an important issue following a long tradition of research on the role of the Action Observation Network (AON) in action understanding. As a consequence of motor mirroring, the involvement of the AON in the identification of others' actions has been assumed to be specific to human actions. The present study demonstrates that some regions of the AON, in particular the lateral occipito-temporal cortex, code for the perception of action events that generalize across animate (human) and inanimate (object) agents.

Multivariate pattern analysis techniques are applied on fMRI data from 25 participants. After observing the brain regions activated during the perception of each type of event, different levels of cross-decoding procedures are conducted to identify the brain regions that generalize over animacy and stimulus modality. The experiment design is elegant and the study rigorously conducted. My comments and suggestions mostly aim at stressing the novelty of the findings and delineate their scope. I also question a few methodological choices. I detail my comments below.

Comment: The study should be better articulated with the theory developed by Jeff Zacks on Event Perception and Representation, which also aims at generalizing across different types or formats of events. A recent review paper is briefly cited on page 1 of the Introduction, but I think that the theoretical issues raised by Zacks et al. and the empirical evidence provided by their work are highly relevant for the current study and deserve more consideration. The similarities and divergences with this line of research should be clearly highlighted throughout the manuscript.

Response: We thank the Reviewer for their kind comments, and for encouraging us to provide a more in-depth discussion of the long tradition of research on event cognition. Based on these comments, we now updated the introduction by contextualizing our study in relation to the research on event cognition, as well as the theory developed by Zacks and colleagues. See the below paragraph for the relevant changes in the Introduction:

“There is a long tradition of research on structured event representations (i.e., event models) that enable predictive processing of complex naturalistic stimuli (for reviews, see Radvansky & Zacks, 2011; Richmond & Zacks, 2017; Zacks, 2020; for a recent computational model, see Franklin et al., 2020). This work revealed neural representations of events that are shared across perception, memory, and language (see Kurby & Zacks, 2008; Speer, Zacks, & Reynolds, 2007; Speer et al., 2009; Zacks, Speer, & Reynolds, 2009), which presumably could capture the shared aspects of actions and object events as well. However, previous work in this domain mostly focused on

actions of humans, especially in the context of how the brain segments ongoing human activity into meaningful elements. Thus, a general neural representation of events that can capture both actions and object events has not been addressed explicitly.” (p. 2)

We also incorporate a discussion of the relevant theoretical and empirical work on event cognition in the Discussion section of our manuscript (p.14). Since the Reviewer raises important issues in their next comment that are also related to the theoretical framework by Zacks and colleagues, we address the changes in the Discussion section under the next comment.

Comment: The events and the actions investigated may be mostly differentiated based on motion trajectories. I understand that this is related to the need to have similar structures in the design of animate/inanimate events, but I believe this is an important specificity to consider and discuss. Should we expect the same degree/locus of common neural substrate for other types of events that do not involve that much motion (e.g., multistep functional events, human using objects versus automatic machine)? For example, one may expect predictive processes to be involved in the understanding of motion events. Variations in predictability of the different motion trajectories might explain the high degree of overlap in the neural substrate of event perception in the animate and inanimate conditions (especially when the task, via the catch-trials, orients towards prediction).

Response:

We agree with the Reviewer that the motion trajectory was the main component that varied across the three events. In future work, we want to address a wider range of movements, potentially ones that are not directly defined by their motion trajectory (e.g., change of state, manner of motion events) to further address the scope of the cross-animacy representations we uncovered. We also agree that the predictability of movement might have played a role in the shared neural representations we identified across actions and object events. As the Reviewer points out, due to the small number of classes we used, and the decoding approach we followed, it was important for us to use actions and object events that are equivalent in their predictability and complexity. If we were to use complex, multistep functional events, encoding and prediction of motion trajectory would not be as key, and motion paths would not be as predictable. Furthermore, as the event becomes more complex, parsing and identifying boundaries of events would become relevant and additional mechanisms would be at play.

In this experiment, we focused on the internal structure of small event chunks. Part of the reason why we focused on such simple event chunks is that the more complex an event gets, the harder it is to find object events that do not give a sense of intentionality or animacy. Human action consists of nested units of events that are defined around goals and intentions. By their very nature, physical events involving objects would be less likely to have such a structure. One

could think of certain sequences of physical events that would be embedded in a multistep hierarchy (e.g., the different phases of a thunderstorm), but finding complex multi-step events that are structurally similar across agents and objects is an experimental challenge in and on itself. Yet, we think that insofar as there is a common event structure across movements of animate and inanimate entities, cross-animacy generalization should be possible. And as the Reviewer implies, the neural loci could be different.

For instance, past neuroimaging work on event cognition primarily used complex, multistep events that cannot be fully defined by motion trajectories. In those studies, event representations that generalize across stimulus formats or specific contexts tend to converge in posterior medial network and hippocampus (Hasson et al., 2015; Kurby & Zacks, 2018; Ranganath & Ritchey, 2012), although effects can be observed in parts of the AON (e.g., posterior temporal cortex, inferior parietal lobe). There is also evidence that different brain regions capture information about events at varying timescales (Baldassano et al., 2017), and unlike more complex multistep events, the events we used in this study are defined at a very short timescale. If we used complex multistep events to capture the shared neural representations of actions and object events, our effects could have centered around these regions.

To address these issues, we now updated parts of the discussion, and hope that the following paragraph better addresses the issues the Reviewer is raising.

“In addition, even though we think that common physics and spatiotemporal characteristics of events underlie the shared neural code we have uncovered, we would like to note that we have tested a small set of events that were primarily defined by their motion trajectories. This limits the conclusions we can derive regarding the underlying factors of cross-animacy generalization. Most events we perform and encounter in our daily life consists of small units organized in a temporal and spatial hierarchy. Whether the neural locus and degree of generalization across actions and object events will extend to these more complex scenarios remains to be seen. For instance, it has been shown that during passive viewing of daily activities or comprehension of narrative texts, posterior medial network regions (i.e., parahippocampal cortex, angular gyrus, medial prefrontal cortex, posterior cingulate, precuneus) and hippocampus are sensitive to information about event dynamics (Baldassano et al., 2018; Hasson et al., 2015; Kurby & Zacks, 2018; Ranganath & Ritchey, 2012). While these regions capture event dynamics at a longer temporal scale, regions in which we focused here (e.g., posterior temporal cortex, inferior parietal lobe, and ventral premotor cortex) capture event boundaries at shorter temporal scales (see Baldassano et al., 2017). Even though we observed cross-animacy generalization in parts of the posterior medial network (e.g., posterior medial parietal lobe, angular gyrus), we did not observe robust cross-animacy generalization in regions such as parahippocampal cortex or medial prefrontal cortex. Future studies can test events across different timescales and levels of complexity to address shared neural representations of actions and object events in different scenarios.” (pp.13-14)

Comment: The critical role of the common node identified should be further discussed. Is it possible that the areas identified do not have the same critical status for the different conditions, despite involving the same neural network? To illustrate with a putative scenario, verbal stimuli might lead to visual simulation of the verbal content, but this simulation might not be essential for the understanding of the sentence. Left LOTC might be therefore critical for visual processing of events but simply a by-product of the processing of event sentences.

Response:

We agree with the Reviewer that cross-animacy + modality generalization does not guarantee that the regions we identified have the same critical status for verbal and visual stimuli. We also agree that visual imagery could potentially underlie the cross-modal effects we identified. Since Reviewer 2 also raised similar concerns, we are repeating our response below:

Generalization across verbal and visual stimuli could be due to both visual imagery (in the sentence session) or verbalization (in the video session). If this is the case, we would expect stronger imagery in the participant group that started with the video session, and stronger verbalization in the participant group that started with the sentence session. Below is the whole-brain contrast of the decoding maps of the two groups, which revealed no significant differences in left LOTC as shown with the black outline. There was a trend for higher decoding in left STS for participants who received the sentence session first, which would be consistent with stronger verbalization in the video session, but the difference was not significant.

Figure 7. Decoding contrast of cross-animacy + modality based on session order. The black outline indicates areas that showed above-chance cross-animacy + modality decoding in the whole group.

Even though we did not find a clear order effect, it's still possible that cross-animacy + modality decoding was at least partly due to visual imagery. For instance, it is possible that the session order did not cause strong differences in the degree of imagery or verbalization, but cross-animacy + modality generalization could still be underlain by it. If so, the decoded

information in left LOTC would be in a verbal format, accessed by both understanding sentences and verbalizing observed actions, or in a visual format, accessed by observing action scenes and visual imagery in the sentence session. This definitely is a possibility, but our work is silent as to the debates regarding the critical role of the common node we identified, and only concern the content of what is being represented. Since functional neuroimaging is necessarily correlational, evidence from individuals with brain damage or neuromodulation studies could better address the critical role of the common node we identified.

To address concerns related to imagery or verbalization that were also partly raised by Reviewer 3, we now added the following paragraph to our manuscript, and report the analysis of the order effect in our supplementary information (see Supplementary Figure 7).

“In the so-called action observation network, only left LOTC/STS showed significant cross-animacy + modality generalization that survived correction for multiple comparisons in the whole brain. Is this region causally involved in understanding events both through visual and verbal formats? If cross-animacy + modality generalization in LOTC is due to verbalization (in the video session) or visual imagery (in the sentence session), this might not be the case. For instance, if generalization across observed videos and sentences is due to imagery in the sentence session, this would not guarantee that neural activity during sentence processing is essential for understanding events in that modality. Since fMRI can only provide correlational evidence, evidence from individuals with brain damage or neuromodulation studies can be used to address the critical role of this region in understanding events through visual and verbal modalities. Still, our study allowed us to address the extent of imagery or verbalization effects as we balanced the order of video and sentence sessions across participants. If across modality generalization is due to imagery, we would expect stronger imagery for people who received the videos first and sentences second, and stronger verbalization for people who received sentences first and videos second. The contrast of the decoding maps of the two groups, however, revealed no significant differences indicative of an effect of imagery or verbalization (see Supplementary Figure 7). Thus, we found no support for the hypothesis that visual imagery or verbalization can fully account for the observed generalization across animacy and modality.” (p.11)

Comment: The analyses logically focus on the “core regions of the action observation network”. Yet ROI selection is not sufficiently justified. On page 16 section ROI Analysis, the authors explain that selection was based on the very nice meta-analysis by Casper et al. (2010) but the regions identified as part of the AON in Casper et al. have not all been shortlisted. In particular, I am very surprised that the ROI analyses did not consider SPL and pMTG. This is even more puzzling that SPL and STS are initially highlighted in the distinct neural representations of actions and object events on page 6. In parallel, I encourage the authors to use a very precise and consistent definition of the AON throughout the paper, as the term does not mean much otherwise.

Response: As the Reviewer points out, we focused on three main regions of the action observation network that are most strongly and consistently activated in action perception studies: lateral occipitotemporal cortex, ventral premotor cortex, and inferior parietal lobe (Caspers et al., 2010; Molenberghs et al., 2012). Lesions to these three main regions are also associated with deficits in action understanding (Urgesi et al., 2014). However, we agree with the Reviewer that it is important for us to clarify and be consistent about what we mean by AON. Based on these concerns that were also raised by Reviewer 2, we now made various changes throughout the manuscript clarifying the definition of the action observation network and what we mean by it. Below are some new paragraphs and sections added to the paper in response to this comment. First, we updated the Introduction to better clarify what we mean by AON, and the regions that we focus.

“Functional neuroimaging has revealed a set of bilateral frontoparietal and posterior temporal regions that are consistently recruited when observing others’ *actions* (e.g., someone jumping). These regions are collectively termed as the *action observation network* (AON), with its frontoparietal component also called the *mirror neuron system* (see Caspers et al., 2010; Kilner et al., 2007; Molenberghs et al., 2012; Watson et al., 2013). With a particular emphasis on lateral occipitotemporal cortex (LOT), inferior parietal lobe (IPL) and ventral premotor cortex (PMv), these regions are thought to play complementary roles in encoding information about observed actions (e.g., Avenanti et al., 2013; Cross et al., 2006; Iacoboni & Dapretto, 2006; Kilner, 2011; Ricciardi et al., 2013; Oosterhof et al., 2013; Urgesi et al., 2014).” (p.2)

Second, we agree with the Reviewer that providing results for SPL and STS will be beneficial, especially since we observe distinct neural representations of actions and object events in these regions. Following the Reviewer’s comments, we now expanded the ROI analysis section of our methods paragraph, justifying the selection of our ROIs, and added the STS and SPL to our analyses. For the STS ROI, we again used the coordinates provided by Caspers et al. (2010). For the SPL ROI, Caspers et al. (2010) provides multiple coordinates, and since we did not have an a priori reason for selecting one over the other, and this region is not a part of the action observation network core, we selected the centroid of the Brodmann area 7 for SPL to maintain consistency across the two hemispheres (TAL coordinates - left STS: [-52,-49,11], left SPL: [-18 - 57 50], right STS: [54, -40, 8], right SPL: [24, -56, 54]). We are now explaining and describing our ROI selection in more detail with the newly edited paragraphs below:

Under Results (p. 6):

“To obtain a better understanding of action and object event decoding across specific regions, we extracted classification accuracies from independently defined regions of interest (ROIs) in each hemisphere. We primarily focused on regions of the action observation network that are most

strongly and consistently recruited during action observation tasks: lateral occipitotemporal cortex (LOTc), ventral premotor cortex (PMv), and inferior parietal lobe (IPL) (for reviews, see Caspers et al., 2010; Molenberghs et al., 2012). To provide a more fine-grained picture of how actions and object events are represented in other areas that are also linked to action observation, we report ROI results from superior parietal lobe (SPL) and superior temporal sulcus (STS) (see Methods for more details on ROI selection).”

Under Methods (p. 18)

“We selected the relevant coordinates based on a meta-analysis of action observation, which revealed increased activity across a range of frontal, temporal, and parietal brain regions during action observation tasks (Caspers et al., 2010). Among these regions, we primarily focused on a bilateral network of three core regions of the “action observation network”— the lateral occipitotemporal cortex (LOTc), the inferior parietal lobule (IPL) and the ventral premotor cortex (PMv) – that are most strongly and consistently recruited during action observation tasks (for reviews, see Caspers et al., 2010; Molenberghs et al., 2012). In addition to these three core regions of the AON, for a more fine-grained understanding of differences between actions and object events, and cross-animacy generalization in AON more broadly, we also report results from superior parietal lobe and posterior superior temporal sulcus. Since the meta-analysis provided multiple ROIs for superior parietal lobe, for simplicity, we used the centroid of Brodmann areas 7 for left and right visuomotor SPL (see Lacadie et al., 2008).”

Comment: The discussion of the main findings, in particular the left/right asymmetry, may also include the role of the methodology used, which is never addressed. For example, I wonder if there is a relation between the overall amount of activity in a given region/hemisphere and the chance of successful cross-decoding in this region/hemisphere. On a related note, would the result about action/event generalization (p. 7 and Fig 3) be stronger with the addition of a control ROI that does not show generalization across animacy?

Response: As we pointed out in the article, data were demeaned for each multivoxel beta pattern in a searchlight sphere by subtracting the mean beta of the sphere from each beta of the individual voxels. Demeaning was applied to make sure that classifiers do not distinguish actions based on global univariate changes only. This, we think, helps address this issue to some extent. However, as per Reviewer’s request, below is a graph that allows the comparison of overall univariate activity against baseline and cross-animacy decoding in the video session.

Figure 8. Comparing univariate activity against baseline and cross-animacy decoding of events in the video session. Error bars indicate standard error of the mean (SEM), and asterisks indicate FDR-corrected effects: * $p < .05$, ** $p < 0.01$, *** $p < 0.001$, **** $p < 0.0001$.

Looking at the activity against baseline for all events, LOTC in both hemispheres showed the highest activity, followed by SPL and STS. In fact, IPL did not even show overall increases in activity against the baseline, however, it showed strong and robust cross-animacy decoding, even higher than PMv, for instance. The fact that the IPL shows weaker univariate activity but stronger decoding than PMv already suggests that differences in decoding cannot be explained away by differences in univariate activity. It can also be seen that the left and right LOTC regions do not show differences in univariate activity strength but left LOTC shows higher cross-animacy decoding than the right.

A control ROI that doesn't show cross-animacy generalization would be an interesting addition, but we were not sure on which basis we would make this choice. We find cross-animacy generalization of observed events in a wide range of areas. We would like to emphasize that the areas that are not colored in the whole-brain maps are areas that do not show significant cross-

decoding of actions and object events. Still, we think that this is partly related to our answer to a previous comment where we stated that we do not observe robust cross-animacy decoding in some regions that have been implicated in representing event dynamics such as parahippocampal cortex or anterior prefrontal cortex.

Following the Reviewer’s request for an additional ROI that doesn’t show cross-animacy decoding, and in relation to this previous point we raised, we performed an additional ROI analysis in anterior prefrontal cortex and parahippocampal cortex to address the Reviewer’s request. Since we did not have a reference point to select these ROIs, we used the centroid of Brodmann areas 10 (anterior prefrontal cortex) and 36 (parahippocampal cortex) for these regions. As it can be seen in the figure below, parahippocampal and anterior prefrontal cortex in both hemispheres showed much lower cross-animacy generalization compared to the regions of the action observation network. In the whole brain, neither cluster survives corrections for multiple comparisons. In the ROI analysis, the parahippocampal cortex shows above chance cross-animacy generalization in both hemispheres, but the anterior prefrontal cortex shows above chance decoding only in the left hemisphere.

Figure 9. Additional ROI analyses of cross-animacy decoding of events in the video session

demonstrating parahippocampal and anterior prefrontal cortices. Error bars indicate standard error of the mean (SEM), and asterisks indicate FDR-corrected effects: * $p < .05$, ** $p < 0.01$, *** $p < 0.001$, **** $p < 0.0001$.

Minor suggestions:

Comment: Presentation of ROI results (p. 6): the presentation of the different contrasts is difficult to follow. I recommend decomposing the interaction in the same manner for left and right hemispheres.

Response: We thank the Reviewer for raising this issue. We now updated the presentation of results in this section and reported the same comparisons for left and right hemisphere ROIs. Note that these outputs now include STS and SPL as well. We also made several changes explaining the logic behind these analyses throughout this section. We hope that the ROI results are now easier to follow. See changes in pp. 6-8.

Comment: Sample size (p. 12): I know that there is no actual real standard for apriori power analyses in this type of neuroimaging study, but I think it is still good practice to justify the sample size considered with the best that can be found.

Response: In response to this comment, we now added the following paragraph under the Participants section of our Methods:

“No statistical test was used to predetermine sample size. We collected data from $N=25$ participants who attended both the video and sentence sessions. Our previous work on human action observation showed that depending on region of interest, actions presented in videos can be decoded with sample sizes between $N = 5$ (left LOTC; with $d = 1.94$, $\alpha = 0.05$, power = 0.95) and $N = 14$ (left PMC; with $d = 0.95$, $\alpha = 0.05$, power = 0.95, see Wurm et al., 2017). Another recent study from our lab showed cross-decoding of human actions across verbal and visual stimuli with $N = 22$ participants (Wurm & Caramazza, 2019). Given these considerations, we think that our sample size of $N = 25$ gives us sufficient power to identify where in the brain information about events are encoded and are in the range of sample sizes conventionally used in the field.” (p. 15)

Comment: Stimuli (p.13): I encourage the authors to provide an illustration of the different possible exemplars for one event as supplementary materials so that the reader can appreciate their variability.

Response: The exemplars of the hit event for actions and object events are now provided in supplementary files on Open Science Framework (<https://osf.io/h4mtp/files/osfstorage>).

Comment: Figure captions: I do not see what is “outlined in black” on the maps.

Response: There was a thin black line surrounding the corrected maps, but the Reviewer is right that they are not very visible, so mentioning them is not very helpful. We removed this information from the captions for Figures 2a-b, and 3 where this information is not very helpful.

Comment: Catch-trials (p. 14): the choice of catch-trials is very important because it defines the task participants actually perform. Were there neural differences between the conceptual vs. perceptual catch-trials? I could not figure out how the different catch trials were organized within blocks/sessions.

Response: We thank the reviewer for encouraging us to provide this information. We now added a sentence in the manuscript clarifying how the different types of catch trials were presented (p. 16). To iterate this information here, we did not split the different catch trial types in different blocks or runs. We compiled all catch trials, shuffled them, and randomly distributed them across runs. That is why we cannot reliably compare univariate activity for different types of catch trials because they were presented all random, and the appearances of perceptual or conceptual catch trials were not controlled to avoid prediction. Furthermore, since catch trials were distributed randomly across all trials, we do not expect them to have a systematic effect on regular trials. Thus, even though it would be interesting to compare neural responses to perceptual versus conceptual catch trials, they are not directly relevant to the results presented in the paper.

Below are the results of the average univariate contrast map of catch versus regular trials for sentence and video sessions to show overall changes in activity during a catch trial versus a regular trial. For simplicity, we’re displaying the left hemisphere results, but similar effects were observed in the right hemisphere.

Figure 10. FDR-corrected univariate contrast map of catch versus regular trials for videos and sentences.

Comments: Event classification (p. 15): I had a hard time understanding to which conditions the two predictors “based on 4 trials selected from the odd and 4 selected from the even trials of a run” refer to. Is it the verbal/ visual modality?

Response: To increase samples per event type for the decoding analyses, we created two predictors per run for each event, both in the verbal and visual modality. The odd appearances of one event were used to generate one predictor, and the even appearances of the same event were used to create the other predictor. This was just done to increase the sample size for decoding analyses. We understand that this might be confusing for the readers and clarified this part of the manuscript (p. 17).

Comment: ROI analysis (p.17): Is it possible to add random slopes for items in the mixed-models used?

Response: We are not sure what the ‘item’ would refer to here since the dependent variable in these analyses is decoding accuracy. Since we are reporting the classification accuracy of the three events, the different items are combined in this analysis.

References provided by the Reviewer

- Zacks J. M. (2020). Event Perception and Memory. *Annual review of psychology*, 71, 165–191. <https://doi.org/10.1146/annurev-psych-010419-051101>
- Radvansky, G. A., & Zacks, J. M. (2011). Event perception. *Wiley interdisciplinary reviews. Cognitive science*, 2(6), 608–620. <https://doi.org/10.1002/wcs.133>

Franklin, N. T., Norman, K. A., Ranganath, C., Zacks, J. M., & Gershman, S. J. (2020). Structured Event Memory: A neuro-symbolic model of event cognition. *Psychological review*, 127(3), 327–361. <https://doi.org/10.1037/rev0000177>

Richmond, L. L., & Zacks, J. M. (2017). Constructing Experience: Event Models from Perception to Action. *Trends in cognitive sciences*, 21(12), 962–980. <https://doi.org/10.1016/j.tics.2017.08.005>

Response: We thank the reviewer for providing us with these papers. We now went over all of them and provided citations and discussions of these papers and the related literature as needed.

Reviewers' Comments:

Reviewer #1:

Remarks to the Author:

The Authors addressed my comments satisfactorily. I read the replies to the other Reviewers' comments and was impressed by the changes made. I see no more issues to be addressed.

Reviewer #2:

Remarks to the Author:

Re-review of Akbiyik

The authors have done a valiant job responding to critiques, sometimes convincingly, and sometimes less so. I still find the evidence weak for the key claim in the paper, as detailed below but given the enthusiasm of the two other reviewers I am prepared to be over ruled.

1. The key claim in the paper is that regions previously implicated in animate action understanding are also engaged in inanimate object actions. One important prediction of this hypothesis is that action decoding will be significant across animacy and modality. The authors find this effect weakly but significantly in Figure 4B of the paper. But that finding is only meaningful if it is also possible to find significant action decoding across modality within objects. These data were not shown before but are now presented in Figure 5 of the reply. (Yes, that is what I intended to request in my earlier review, I am sorry I wrote it wrong but the authors correctly inferred what I meant.) But what Figure 5F of the reply shows is that none of the ROIs show significant cross-modal decoding of object events. That seems to me quite problematic for the main claim in the paper, even given the cross-modal and cross-animacy decoding.

2. The issues with retinotopic cortex are less serious but I'll note them briefly.

I appreciate that the authors have analyzed early visual cortex to see if indeed action decoding can be found there. It seems suboptimal to use the huge anatomical ROIs defined by Tal coordinates with a 12 mm radius (!), rather than using sulcus-based atlases that accurately define V1. This could reduce the authors' power to detect cross-modal decoding in V1.

Further, I disagree with the authors' argument that: "it is not clear how this raises a theoretical concern for our interpretations. First, let us assume that cross-animacy decoding in occipital lobes is driven solely by variation in low-level visual properties. This does not necessarily mean that decoding in frontoparietal regions and temporal cortex more anteriorly also reflect encoding of such properties. The possibility that decoding in early visual regions might capture variation in low-level visual features does not explain why we observe decoding in higher-level brain regions".

But parsimony dictates that you don't get to make the fancy new high-level claim if the boring old low-level claim suffices. Decoding in V1 indicates that low-level confounds are present, and if they are, it is very hard to rule them out for higher-level regions. It is not that this proves the effects in higher-level regions are necessarily due to those confounds. But the burden is on the authors to rule out this account, and I don't see that they do.

Reviewer #3:

Remarks to the Author:

The Authors have made a thorough revision of their manuscript. The additional information and analyses provided in the revised document answered all my previous questions. I do not have other comments and I recommend publication of the manuscript in its current form.

Responses to Reviewer's Comments

Reviewer #1 (Remarks to the Author): The Authors addressed my comments satisfactorily. I read the replies to the other Reviewers' comments and was impressed by the changes made. I see no more issues to be addressed.

Reviewer #3 (Remarks to the Author): The Authors have made a thorough revision of their manuscript. The additional information and analyses provided in the revised document answered all my previous questions. I do not have other comments and I recommend publication of the manuscript in its current form.

Response: We thank the Reviewers 1 & 3 for their kind comments and are happy to see that we addressed their comments satisfactorily and that they're happy with the publication of the article as is. We believe that their comments added a lot of value to our paper, and we appreciate their time and effort.

Reviewer #2 (Remarks to the Author): The authors have done a valiant job responding to critiques, sometimes convincingly, and sometimes less so. I still find the evidence weak for the key claim in the paper, as detailed below but given the enthusiasm of the two other reviewers I am prepared to be overruled.

1. The key claim in the paper is that regions previously implicated in animate action understanding are also engaged in inanimate object actions. One important prediction of this hypothesis is that action decoding will be significant across animacy and modality. The authors find this effect weakly but significantly in Figure 4B of the paper. But that finding is only meaningful if it is also possible to find significant action decoding across modality within objects. These data were not shown before but are now presented in Figure 5 of the reply. (Yes, that is what I intended to request in my earlier review, I am sorry I wrote it wrong, but the authors correctly inferred what I meant.) But what Figure 5F of the reply shows is that none of the ROIs show significant cross-modal decoding of object events. That seems to me quite problematic for the main claim in the paper, even given the cross-modal and cross-animacy decoding.
2. The issues with retinotopic cortex are less serious but I'll note them briefly. I appreciate that the authors have analyzed early visual cortex to see if indeed action decoding can be found there. It seems suboptimal to use the huge anatomical ROIs defined by Tal coordinates with a 12 mm radius (!), rather than using sulcus-based atlases that accurately define V1. This could reduce the authors' power to detect cross-modal decoding in V1.

Further, I disagree with the authors' argument that: "it is not clear how this raises a theoretical concern for our interpretations. First, let us assume that cross-animacy decoding in occipital lobes is driven solely by variation in low-level visual properties. This does not necessarily mean that decoding in frontoparietal regions and temporal cortex more anteriorly also reflect encoding of such properties. The possibility that decoding in early visual regions might capture variation in low-level visual features does not explain why we observe decoding in higher-level brain regions". But parsimony dictates that you don't get to make the fancy new high-level claim if the boring old low-level claim suffices. Decoding in V1 indicates that low-level confounds are present, and if they are, it is very hard to rule them out for

higher-level regions. It is not that this proves the effects in higher-level regions are necessarily due to those confounds. But the burden is on the authors to rule out this account, and I don't see that they do. The authors have done a valiant job responding to critiques, sometimes convincingly, and sometimes less so. I still find the evidence weak for the key claim in the paper, as detailed below but given the enthusiasm of the two other reviewers I am prepared to be overruled.

Overall response: We thank the Reviewer for their time and effort. However, we think that there is a mismatch in our analytical and theoretical approach and what Reviewer 2 counts as preconditions for the validity of our findings. Both in their original review and in their current comments, the Reviewer is framing our findings and interpretation in a way that is not consistent with how we frame them.

In our paper, we provided evidence from both visual and verbal modalities, which address different aspects of the neural representation of actions and events. We think that it is important to evaluate these different analytical approaches (especially the analyses on visual and verbal modalities) on their own merits to better judge their implications. Furthermore, the Reviewer stated that we addressed their previous comments sometimes *'convincingly and sometimes less so'* but did not clearly explain what they mean by this or provide suggestions for improvement. Thus, it wasn't clear to us what revisions would alleviate their concerns. Yet, as detailed below, we tried to address their points as much as possible and updated the manuscript accordingly. Since they nicely stated that they *"... [are] happy to be overruled given the enthusiasm of the other Reviewers"*, we hope that the paper won't be sent back for review and will be accepted in its current form. Below is our point-by-point response to their comments.

Comment 1 (first half): *"The key claim in the paper is that regions previously implicated in animate action understanding are also engaged in inanimate object actions. One important prediction of this hypothesis is that action decoding will be significant across animacy and modality."*

The first sentence accurately reflects the intended and stated key claim in the paper. This claim – *"that regions previously implicated in animate action understanding are also engaged in inanimate object actions."* – rests primarily on our action and event observation experiment with visual stimuli of actions and events. However, the second sentence is inaccurate. A possible cross-animacy + modality generalization is not a necessary prediction of a common representation of actions and object events shown in the visual modality. A region could be encoding information about the shared visual and spatiotemporal characteristics of movement across actions and object events, but it may not capture their shared linguistic representations. Absence of visual to linguistic generalization in that region would not raise doubts about generalization observed within the visual modality. Once a generalization across agent actions and object events is established in the visual modality, additional questions can be asked about the abstractness of these representations. In this vein, we also asked whether there are regions that capture abstract event information that generalizes across both actions and object events and visual and linguistic modalities. Here, the results pointed to a limited set of brain regions, for instance the left LOTC. Following the reviewer's comment, we have made wording changes to further ensure the clarity of the theoretical framework of our research.

Comment 1 (second half): “But that finding (cross-animacy + modality decoding) is only meaningful if it is also possible to find significant action decoding across modality within objects. These data were not shown before but are now presented in Figure 5 of the reply. (Yes, that is what I intended to request in my earlier review, I am sorry I wrote it wrong, but the authors correctly inferred what I meant.) But what Figure 5F of the reply shows is that none of the ROIs show significant cross-modal decoding of object events. That seems to me quite problematic for the main claim in the paper, even given the cross-modal and cross-animacy decoding.”

Here, the Reviewer suggests that *cross-animacy + modality decoding would only be meaningful if we found cross-modality decoding within the object event condition*. Since we do not find strong effects in the cross-modality decoding of object events, and that this effect seems to be weaker than cross-animacy + modality generalization, the Reviewer thinks that the main claim of the paper is not valid. We had already pointed out the weaker cross-modal decoding of object events in Supplemental Figure 6 provided as part of the revision in response to the first round of reviews. The weaker cross-modality decoding of object events, we think, can be directly linked to noisier data for object events in the sentence session. There could be various reasons for this that we had already addressed in our Supplemental Figure 6. In response to the Reviewer’s current comments, we revised our Supplemental Figure 6 to address this issue in more detail (see the highlighted portions of the Supplemental Information document). However, to reiterate, the main claim of the paper is that the brain regions that have previously been implicated in human action recognition encode a more general representation that captures both observed human actions and object events. We argue that this general representation captures the common physics and structure of events and address this core question in the context of the visual modality. Even if we hadn’t had any data on sentence processing, this claim would still be valid given the robust evidence in the visual modality.

Comment 2: “The issues with retinotopic cortex are less serious but I’ll note them briefly. I appreciate that the authors have analyzed early visual cortex to see if indeed action decoding can be found there. It seems suboptimal to use the huge anatomical ROIs defined by Tal coordinates with a 12 mm radius (!), rather than using sulcus-based atlases (!) that accurately define V1. This could reduce the authors’ power to detect cross-modal decoding in V1. Parsimony dictates that you don’t get to make the fancy new high-level claim if the boring old low-level claim suffices. Decoding in V1 indicates that low-level confounds are present, and if they are, it is very hard to rule them out for higher-level regions. It is not that this proves the effects in higher-level regions are necessarily due to those confounds. But the burden is on the authors to rule out this account, and I don’t see that they do.”

It’s not clear what specific analysis the Reviewer is referring to with this comment. In their original review, they had requested that we look at primary visual cortex for both visual and cross-modal analyses. In the video conditions, we found effects in V1 while we didn’t find effects in V1 for cross-modal decoding. It is important to highlight that the Reviewer’s notion of *visual confounds* is not clear since the implications of visual features for action videos and action sentences are different. For videos, shared visual features of events are not necessarily confounds. They are part of what defines the spatiotemporal characteristics of a dynamic scene. That there are these shared visual features between human actions and object events is literally the insight that underlies the paper. *Bouncing* or *jumping* indicate horizontal movement while *rolling* or *walking* do not. This does not mean that horizontal movement is a visual confound: such visual

information is one aspect of how we distinguish observed events, and it may well be captured by neural activity in early visual cortex.

If the Reviewer instead is talking about the potential visual confounds for cross-modal decoding, we did not find effects in V1 in this analysis. In response to this lack of decoding in V1, the Reviewer raises issues with our ROI selection. We used the centroid of the Brodmann area to perform the analyses since we did not conduct retinotopic mapping. We're aware that the 12mm radius is not small, but it is the same radius as the one we used for all ROIs, so we followed the same ROI size for fair comparison. Nonetheless, to address the point the Reviewer is raising about the radius, below is a graph reporting the same ROI results, now with a much smaller, 6mm ROI.

As can be seen from this graph, we still don't find significant cross-animacy + modality generalization in primary visual cortex. The Reviewer might be right that one could find an effect with a different ROI definition. However, as we pointed out in our original response, even if we did find an effect, we don't think this analysis adds much value for evaluating the evidence presented in the paper regarding the putative action recognition network. The Reviewer argues that this analysis is valuable by appealing to the notion of parsimony. However, appeal to parsimony is only relevant when comparing accounts with the same explanatory power: appealing to a notion of differences among responses in early visual cortex as an explanation of the effects observed in the action recognition network is hardly explanatory.